# MMTU: A Massive Multi-Task Table Understanding and Reasoning Benchmark

**Junjie Xing**[*]
University of Michigan

**Yeye He**[†]
Microsoft Corporation

**Mengyu Zhou**
Microsoft Corporation

**Haoyu Dong**
Microsoft Corporation

**Shi Han**
Microsoft Corporation

**Lingjiao Chen**
Microsoft Corporation

**Dongmei Zhang**
Microsoft Corporation

**Surajit Chaudhuri**
Microsoft Corporation

**H. V. Jagadish**
University of Michigan

## Abstract

Tables and table-based use cases play a crucial role in many important real-world applications, such as spreadsheets, databases, and computational notebooks, which traditionally require expert-level users like data engineers, data analysts, and database administrators to operate. Although LLMs have shown remarkable progress in working with tables (e.g., in spreadsheet and database copilot scenarios), comprehensive benchmarking of such capabilities remains limited. In contrast to an extensive and growing list of NLP benchmarks, evaluations of table-related tasks are scarce, and narrowly focus on tasks like NL-to-SQL and Table-QA, overlooking the broader spectrum of real-world tasks that professional users face. This gap limits our understanding and model progress in this important area.

In this work, we introduce MMTU, a large-scale benchmark with around 28K questions across 25 real-world table tasks, designed to comprehensively evaluate models ability to understand, reason, and manipulate real tables at the expert-level. These tasks are drawn from decades' worth of computer science research on tabular data, with a focus on complex table tasks faced by professional users. We show that MMTU require a combination of skills – including table understanding, reasoning, and coding – that remain challenging for today's frontier models, where even frontier reasoning models like OpenAI GPT-5 and DeepSeek R1 score only around 69% and 57% respectively, suggesting significant room for improvement. We highlight key findings in our evaluation using MMTU and hope that this benchmark drives further advances in understanding and developing foundation models for structured data processing and analysis. Our code and data are available at `https://github.com/MMTU-Benchmark/MMTU` and `https://huggingface.co/datasets/MMTU-benchmark/MMTU`.

## 1 Introduction

Remarkable progress has been made in foundation models [43, 44, 72, 121, 37], partly thanks to an expanding array of large-scale benchmarking efforts. Prominent examples include benchmarks for general language understanding (e.g., GLUE [123], Super-GLUE [124], BIG-Bench [117], MMLU [78], MMLU-pro [127]), as well as benchmarks focused on coding and STEM reasoning

---

[1]Correspondence: jjxing@umich.edu

[2]Correspondence: yeyehe@microsoft.com

39th Conference on Neural Information Processing Systems (NeurIPS 2025) Track on Datasets and Benchmarks.

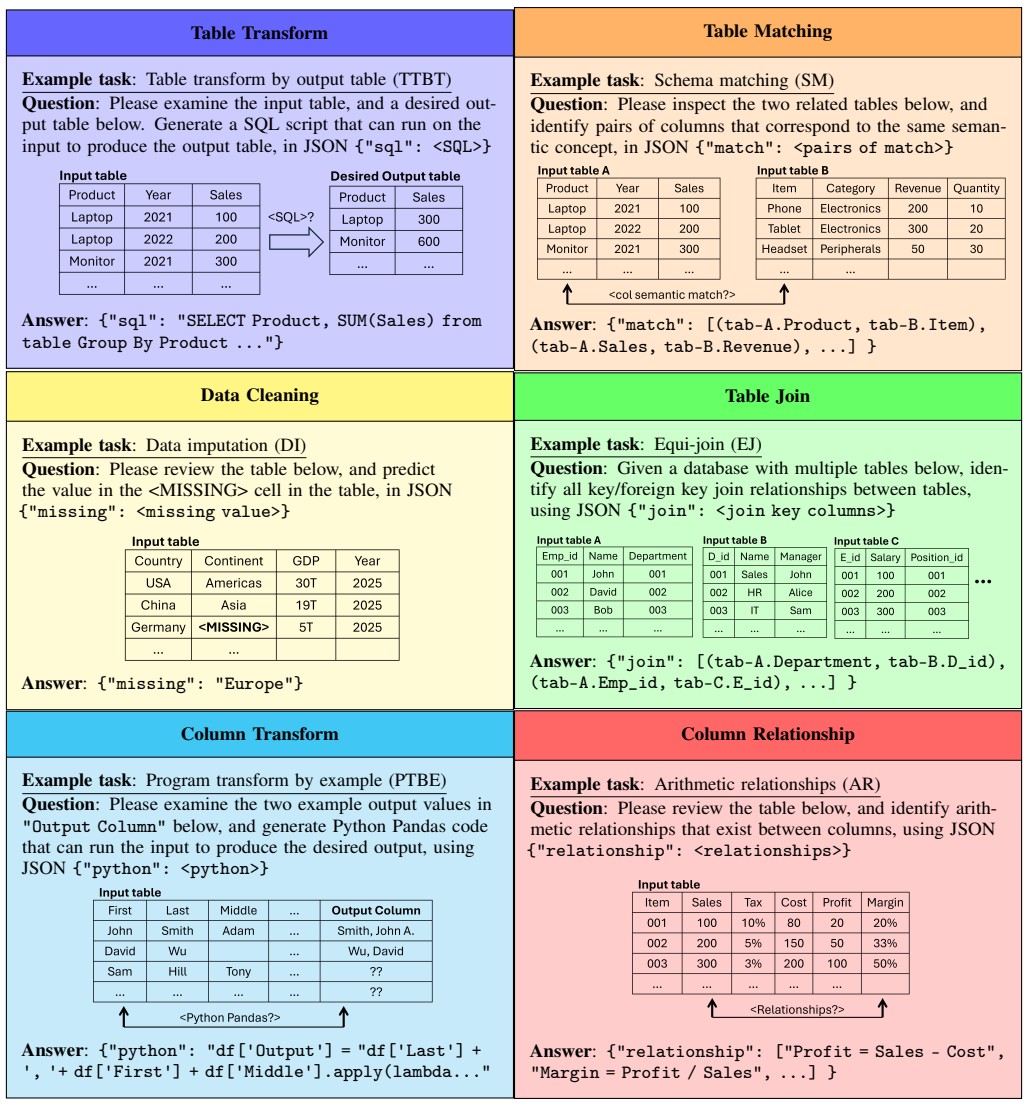

Figure 1: Example questions sampled from different task categories in MMTU for illustrative purposes. Note that questions in MMTU all follow a standardized triplet format of: (Instruction, Input-table(s), Ground-truth answer). The list of all tasks can be seen in Table 1.

(e.g., SWE-bench [13], GPQA-diamond [109], AIME [5], LiveCodeBench [11]). These efforts have played a critical role in deepening our understanding and accelerating the progress of foundation models.

Tables and table-based use cases are central to many real-world applications, including spreadsheets [21, 2], databases [17, 18], and computational notebooks [25, 19], which often require expert-level users such as data engineers, analysts, and database administrators to operate. While LLMs have shown great promise in working with tables [22, 16, 24], existing evaluations of table-tasks remain narrow in scope – primarily focusing on NL-to-SQL [92, 147, 138] and Table-QA [137, 58, 53] – and fail to reflect the broader spectrum of real-world tasks that professional users face.

In this work, we introduce MMTU, a large-scale and challenging benchmark comprising 28,136 questions across 25 real-world table tasks. It is designed to comprehensively evaluate models' ability to understand, reason, and manipulate real tables at the expert-level. The tasks we collect in MMTU are drawn from decades of computer science research on tabular data, with a particular focus on complex tasks that professional users face, as illustrated in Figure 1.

Our evaluation shows that the complex and technical nature of MMTU demands a combination of capabilities – including table understanding, reasoning, and coding – that remain challenging for

Table 1: Tasks and datasets in the MMTU benchmark. Most of these tasks have not traditionally been used to evaluate foundation models (except NL-to-code, Table QA, and KB mapping).

| Task Category | Task Name | Task Description | Metric | References and Datasets | # Questions |
|---|---|---|---|---|---|
| Table Transform | table-transform-by-relationalization (TTBR) | Relationalize a table using transformation | Acc | [93, 39, 83, 132, 87] | 230 |
| | table-transform-by-output-schema (TTBS) | Synthesize transformation by output schema | Acc | [135, 113, 114] | 685 |
| | table-transform-by-output-table (TTBT) | Synthesize transformation by input/output tables | Acc | [40, 125, 122, 142] | 86 |
| Table Matching | Entity matching (EM) | Match rows refer to the same semantic entity | Acc | [65, 103, 146, 95, 105] | 4228 |
| | Schema matching (SM) | Match columns refer to the same concept | F1 | [86, 145, 41, 100] | 669 |
| | Head value matching (HVM) | Match column-headers with cell-values | Acc | [94, 108] | 900 |
| Data Cleaning | data-imputation (DI) | Predict missing values in tables | Acc | [45, 68, 94] | 1971 |
| | error-detection (ED) | Detect erroneous cells in tables | F1 | [60, 126, 101, 50] | 1891 |
| | list-to-table (L2T) | Split lists of undelimited values into table | Acc | [59, 46, 69, 90] | 957 |
| Table Join | semantic-join (SJ) | Predict semantic join between two tables | Acc | [75, 35, 64, 128] | 130 |
| | equi-join-detect (EJ) | Predict equi-joins between a set of tables | F1 | [96, 56, 111, 81] | 494 |
| Column Transform | program-transform-by-example (PTBE) | Program transformation by input/output examples | Acc | [76, 70, 67, 116, 77] | 558 |
| | formula-by-context (FBC) | Predict formula based on table context | Acc | [52, 54] | 3513 |
| | semantic-transform-by-example (STBE) | Predict semantic transformations by examples | Acc | [75, 35, 64] | 131 |
| Column Relationship | arithmetic-relationship (AR) | Predict arithmetic-relationship (AR) in tables | F1 | [115, 1] | 818 |
| | functional-relationship (FR) | Predict functional-relationship (FR) in tables | F1 | [1, 106, 129] | 267 |
| | string-relationship (SR) | Predict string-relationship (SR) in tables | F1 | [1, 76, 71] | 765 |
| Table understanding | Needle-in-a-haystack-table (NIHT) | Retrieve cell content in a table | Acc | [68] | 999 |
| | Needle-in-a-haystack-index (NIHI) | Retrieve index based on cell value | Acc | [68] | 1000 |
| NL-2-code | NL-2-SQL (NS) | Translate natural-language into SQL | Acc | [147, 92, 89, 138, 148] | 2489 |
| Table QA | Table Question Answering (TQA) | Answer questions based on tables | Acc | [131, 137, 57, 58] | 1793 |
| | Fact Verification (FV) | Verify facts based on tables | Acc | [53, 136, 141] | 916 |
| KB Mapping | Column type annotation (CTA) | Predict KB types based on column content | Acc | [82, 134, 118, 140, 80] | 881 |
| | Column property annotation (CPA) | Predict KB property for a pair of columns | Acc | [82, 63, 104] | 873 |
| | Cell entity annotation (CEA) | Predict KB entity for a table cell | Acc | [82, 66, 42, 130, 98] | 892 |
| Total | | | | | 28,136 |

today's frontier models. Even the top performing models, such as OpenAI GPT-5 and DeepSeek-R1 , achieve only 69.6% and 57.9% on MMTU, suggesting substantial room for improvement, and highlighting MMTU as a strong testbed for models aspiring toward general human-level intelligence, e.g., to meet or surpass the top 10% of skilled adults in diverse technical tasks like those characterized in [102].

We perform extensive experiments benchmarking a large collection of models using MMTU, and performed extensive analysis. Some of the key findings from our evaluation include:

- LLMs demonstrate strong potential in understanding and manipulating tabular data. Newer and larger models substantially outperform older and smaller ones, indicating significant advancements in table-related capabilities as captured by MMTU.
- Reasoning models, such as OpenAI GPT-5 and DeepSeek R1, show a clear advantage over general-purpose chat models like GPT-5-Chat and DeepSeek-V3. The top reasoning models outperform the top chat models by over 10 percentage points (Table 3), underscoring the complexity of the tasks (which often require coding in SQL/Pandas) and the importance of reasoning in MMTU.
- Unlike earlier models, today's frontier models are less sensitive to how tables are formatted and serialized (e.g., markdown/CSV/JSON/HTML), reflecting general progress in models' abilities in understanding diverse data formats (Figure 8).
- LLMs still struggle with long table context, or large tables with many rows and columns. Complex tasks requiring holistic reasoning across cell values, especially in the column direction, remains challenging when the table context is long (Figure 6 and Figure 11).
- LLM performance can degrade under table-level perturbations such as row or column shuffling, even when these changes are supposed to be semantically invariant in the context of tables (Figure 7). This sensitivity points to possible limitations in models' ability to understand tables in a robust manner.

We hope MMTU can serve as a valuable addition to the growing landscape of model benchmarks, helping to track progress, identify limitations, and ultimately drive further advancements in this important area of using LLMs for table tasks.

## 2 Related Work

**Large-scale benchmarks for foundation models.** The rapid advancement of foundation models has made benchmark evaluation ever more important. Prominent large-scale benchmarks, such as GLUE [123] (9 NLP tasks), Super-GLUE [124] (10 NLP tasks), Big-BENCH [117] (204 tasks), MMMU [78] (15,908 questions), MMMU-pro [127] (12,032 questions), MMLU [139] (11,550 multi-modal questions) etc., offer comprehensive evaluations of model capabilities. However, as models improve rapidly, benchmarks can become saturated quickly (e.g., in the matter of a few years), prompting newer and more challenging benchmarks [124, 127]. All of these benchmarking efforts

have nevertheless played a crucial role in measuring and stimulating the development of foundation models.

To contribute to this growing landscape, our new MMTU benchmark comprises 28,136 challenging questions across diverse table tasks that expert users would face, which is comparable in scale with prior efforts such as MMMU and MMLU. It complements existing benchmarks, by enabling comprehensive evaluation of foundation models in the important yet underexplored area of table reasoning and understanding.

**Benchmarks for reasoning.** Reasoning has recently emerged as a key challenge for foundation models. Beyond general intelligence benchmarks (e.g., GPQA Diamond [109], AGIEval [149], HLE [8]), there are specialized benchmarks targeting mathematical reasoning (e.g., AIME [5], MathVista [12], IMO [9]), coding (e.g., SWE-bench [13], CodeForce [7], LiveCodeBench [11], IOI [10]), and multi-modal reasoning (e.g., MMMU [139], ARC [6]). These benchmarks have become important tools for evaluating models' ability to tackle complex reasoning tasks, but can also get saturated quickly (e.g., GSM8k [61], Math500 [79], HuamEval [49]), making it necessary to create new and more challenging benchmarks.

We show that our MMTU benchmark can serve to complement existing reasoning benchmarks, by enabling evaluation on complex table-based tasks that require two-dimensional table understanding, coding, and logical reasoning. MMTU extends current reasoning benchmarks into the important yet underexplored domain of tabular data, which underpins many real-world applications.

**Existing benchmarks relating to tables.** Given the importance of table data, benchmarks have been developed in the ML and NLP community to evaluate model ability on tables, which however usually focus on a small set of table tasks such as NL-2-SQL [147, 92, 89, 138, 148] and Table-QA [53, 131, 137, 57, 58]. In contrast, MMTU expands the scope of current evaluations by incorporating 19 additional table tasks drawn from decades of research in communities such as data management and programming languages, resulting in a more comprehensive evaluation framework for assessing LLM capabilities on tabular data.

More recently, spreadsheet-centric benchmarks have emerged, including Spreadsheet-Bench [99], SheetCopilotBench [91], and Sheet-RM [55]. While these are important in the spreadsheet domain, these efforts are typically limited in scale (containing a few hundred cases), and are closely tied to specific file formats (e.g., .xlsx) and software environments. In comparison, MMTU focuses on general-purpose tabular data that applies broadly across spreadsheet, database, and computational notebook settings, enabling more scalable and format-independent evaluation of foundation models.

## 3 MMTU Benchmark for Tables

### 3.1 Benchmark overview

We introduce our Massive Multi-task Table Understanding and Reasoning (MMTU) benchmark, designed to evaluate models table understanding and reasoning capabilities at the expert-level, across a wide range of real-world tasks that would typically be performed by professional data engineers, data scientists, and database administrators.

The benchmark comprises 28,136 complex table-centric questions over 61,763 real tables, in 25 distinct task categories. Each question has a standardized format of "`<Instruction, Input-Table(s), Ground-truth answer>`", like illustrated in example questions in Figure 1. Detailed statistics of the benchmark can be found in Table 2, which highlight the diversity and complexity of the questions in MMTU.

These questions are meticulously collected and curated based on decades of computer science research in areas beyond ML/NLP – such as data management and programming languages – drawing on expert-labeled datasets developed over many years by researchers in these communities, as we will detail below which we will describe below.

### 3.2 Data curation workflow

Figure 2 illustrates the key steps in the overall workflow of our data curation process for producing MMTU. We detail each step in turn below.

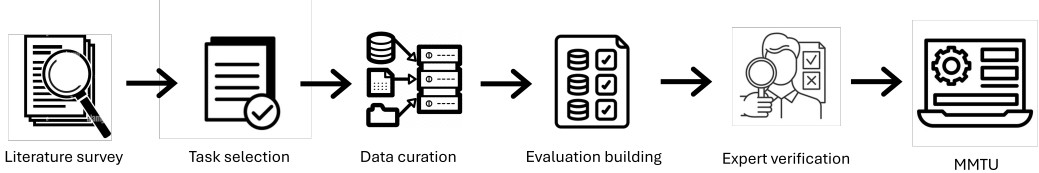

Literature survey → Task selection → Data curation → Evaluation building → Expert verification → MMTU

Figure 2: MMTU data curation workflow: we survey real-world table tasks from the literature, select 25 user-facing tasks with objective evaluation criteria, curate questions from 52 datasets, develop evaluation scripts and verify the results for these tasks, before arriving at the MMTU benchmark.

| Statistics | Number |
|---|---|
| Total Questions | 28,136 |
| Total tasks / datasets | 25/52 |
| Total tables | 61,763 |
| Coding Questions | 7,331 (26.1%) |
| - SQL questions | 2,489 (18.8%) |
| - Python Pandas questions | 1,329 (4.5%) |
| - Spreadsheet Formula questions | 3,513 (11.9%) |
| Non-coding Questions | 20,805 (73.9%) |
| Questions with tables | 28,136 |
| - Questions with 1 table | 20,048 (71.3%) |
| - Questions with 2 tables | 5,285 (18.8%) |
| - Questions with 3 or more tables | 2,803 (10.0%) |
| Table characteristics | |
| - Average table row count | 2,659 |
| - Average table column count | 11 |
| - Average table cell count | 33,251 |
| Table sources | |
| - Web tables | 46,264 (74.9%) |
| - Spreadsheet tables | 4,540 (7.4%) |
| - Relational tables | 10,959 (17.7%) |

Table 2: Statistics of benchmark questions

# Questions by Task Category

Figure 3: Question distribution by task category

**Literature survey.** To ensure our table tasks reflect real-world challenges, we draw on our experience working on related problems, that many challenging predictive table tasks have been studied in the decades worth of computer science research, particularly in data management (SIGMOD/VLDB), programming languages (PLDI/POPL), and web data (WWW/WSDM) communities. We conduct a systematic survey of publications from these venues over the past two decades, leveraging a combination of keyword searches of paper titles, and DeepResearch-like tools [20] (where we specify detailed requirements for papers in these venues), to arrive at a promising set of papers and possible candidate tasks.

**Task selection.** We manually examine candidate tasks described in the surveyed papers from the previous step, and select tasks that are:

(1) Real user-facing tasks, involving data tables that would otherwise require expert-level humans to perform. (We therefore exclude system-level predictive table tasks focused on performance improvements, such as query optimization [85, 88] and cardinality estimation [73, 84]);
(2) Objectively evaluable tasks, that come with unique manually-labeled ground truth. (We therefore exclude tasks such as table summarization [144, 38, 74, 107, 48] and table augmentation [143, 133], which lack unique ground-truth and may require subjective fuzzy LLM-based evaluations);
(3) Tasks based on real-world data tables, which can be real web tables, spreadsheet tables, or relational tables, etc. (We exclude tasks and datasets based on synthetic or perturbed data).
After the selection step, we arrive at 25 different tasks (listed in Table 1) from 52 diverse benchmark datasets (can be seen in Table 5 in the Appendix). We note that a majority of these real-world tasks (all except the last 3 categories in Table 1, NL-2-code, Table-QA and KB-mapping) have not been used to evaluate foundation models. A summary of these tasks is described in more detail in Appendix B.

**Data standardization and curation.** To accommodate the heterogeneity across the 52 benchmark datasets (which have diverse data formats, ground-truth labels, and task definitions), we next standardize the questions in each dataset into a consistent "<Instruction, Table(s), Ground-truth>"

format. Figure 1 shows examples of the triplet format for different tasks. This enables consistent representation across tasks, facilitating the integration of diverse table tasks within a single benchmark framework, and allowing different LLMs to be plugged in for easy model prediction and evaluation.

To ensure the quality of MMTU, we conduct an LLM-based quality check using o4-mini. The model is prompted to evaluate each question for two aspects: (1) whether the question is ambiguous (since tasks like NL-code can sometimes be inherently ambiguous [112]), and (2) whether the ground-truth answer is correct. Approximately 8% of the original questions were flagged as either ambiguous or incorrect and subsequently removed to enhance benchmark quality.

As an additional safeguard, we also use LLM to exclude any benchmark instances that might pose privacy or security risks. To maintain diversity across sources, we cap the number of questions drawn from any single dataset at 1000.

The final composition of the questions is reported in Table 1, with more statistics shown in Table 2 and Figure 3.

**Evaluation framework.** In contrast to benchmarks like MMLU [78] and MMMU [139], which primarily use multiple-choice formats (where evaluation involves comparing a predicted option such as "A/B/C/D" against a single-letter ground truth), real-world table tasks performed by professional experts are often more complex and nuanced. These tasks, such as code generation or structured reasoning, cannot be adequately evaluated using multiple-choice alone. In MMTU, we instead adopt a structured yet open-ended answer format (see Figure 1 for examples) for prediction and evaluation.

To support evaluations beyond simple string comparisons, we designed a lightweight evaluation framework that supports diverse evaluations in table tasks, including execution-based evaluation (for SQL and Python generation), and structured output evaluation (e.g., comparing an unordered JSON list against ground truth). Our framework is also extensible, making it easy to incorporate new task types and evaluation metrics. Details of our evaluation framework can be seen in [15].

**Expert verification.** As a final verification step, we sample 20 questions per task and employ domain experts with years of experience to manually review and verify that (1) raw data is integrated correctly, (2) the ground-truth aligns with human intuition and passes verification, (3) the reference instruction properly reflects the task to produce the desired output, and (4) the evaluation script is set up correctly to correctly evaluate model predictions against ground-truth.

After completing all data curation steps in the workflow illustrated in Figure 2, we obtain a total of 28,136 questions that form our MMTU benchmark, with main statistics summarized in Table 2.

## 3.3 Broader discussions

**Limitations**. A main limitations of our benchmark is how our tasks are sampled and selected. Like discussed in Section 3.2, for ease of evaluation, we include only tasks that can be objectively evaluated, and omit ones that are subjective or creative in nature (e.g., table summarization, generation, and enrichment) that are also important to users, but not included in the benchmark.

In addition, since we sampled table tasks from the existing research literature, which naturally introduces biases, as it omits tasks that are important in practice but not well studied in the literature, or tasks that lack good labeled data (e.g., multi-turn table manipulation).

Lastly, while human experts often read tables visually on two dimensional grid (which makes two dimensional spatial reasoning easy), our current evaluations only use text-based input, and do not consider multi-modal input. Extending the benchmark to multi-modal table input is an interesting direction for future work.

**Broader impacts.** Our benchmark is designed to evaluate LLMs performance on challenging expert-level table tasks, with the goal of identifying model shortcomings and stimulating model improvements. We hope this can lead to more capable models, to better assist human users in scenarios such as spreadsheet-copilot and database-copilot.

We make our best efforts to exclude any content that may raise privacy or security concerns, contain explicit material, depict violence, or be otherwise sensitive. For example, we manually review instructions and datasets at the task level, and employ GPT-4o at the data record level, to remove any that may have privacy concerns, in order to minimize potential negative effect of the data.

| Model Type | Model | MMTU result | Cost per question (US$) |
|---|---|---|---|
| Reasoning | GPT-5 | **0.696 ± 0.01** | 0.01727 |
| | o3 | 0.691 ± 0.01 | 0.01539 |
| | GPT-5-mini | 0.667 ± 0.01 | 0.00276 |
| | Gemini-2.5-pro | 0.665 ± 0.01 | 0.00790 |
| | o4-mini (2024-07-18) | 0.660 ± 0.01 | 0.00993 |
| | Grok-3-mini | 0.645 ± 0.01 | 0.00111 |
| | Gemini 2.5 Flash | 0.625 ± 0.01 | 0.00199 |
| | Deepseek-R1 | 0.579 ± 0.01 | 0.00167 |
| | gpt-oss-120b | 0.543 ± 0.01 | 0.00050 |
| | Qwen3-235B-A22B-Thinking-2507 | 0.529 ± 0.01 | 0.00068 |
| | Qwen3-32B (thinking) | 0.506 ± 0.01 | 0.00017 |
| | gpt-oss-20b | 0.478 ± 0.01 | 0.00040 |
| | Qwen3-8B (thinking) | 0.473 ± 0.01 | 0.00041 |
| Chat | GPT-5-Chat | **0.577 ± 0.01** | 0.00534 |
| | Deepseek-V3 | 0.555 ± 0.01 | 0.00095 |
| | Qwen3-235B-A22B-Instruct-2507 | 0.524 ± 0.01 | 0.00044 |
| | GPT-4o (2024-11-20) | 0.507 ± 0.01 | 0.01019 |
| | Llama-4-Maverick-17B-128E-Instruct-FP8 | 0.490 ± 0.01 | 0.00066 |
| | Llama-3.3-70B | 0.454 ± 0.01 | 0.0015 |
| | Mistral-Large-2411 | 0.446 ± 0.01 | 0.01066 |
| | Mistral-Small-2503 | 0.417 ± 0.01 | 0.00273 |
| | GPT-4o-mini (2024-07-18) | 0.400 ± 0.01 | 0.00061 |
| | Llama-4-Scout-17B-16E-Instruct | 0.393 ± 0.01 | 0.00036 |
| | Qwen3-32B (no thinking) | 0.379 ± 0.01 | 0.00007 |
| | Qwen3-8B (no thinking) | 0.353 ± 0.01 | 0.00015 |
| | Qwen2.5-7B-Instruct | 0.310 ± 0.01 | 0.00010 |
| | Llama-3.1-8B | 0.268 ± 0.01 | 0.00003 |

Table 3: Overall performance and cost results of chat and reasoning models. Cost per question is calculated based on public pricing information on `https://openrouter.ai/` as of Aug 2025. Results for top-performing models, such as GPT-5 and GPT-5-Chat, are averaged over 3 runs.

## 4 Experiments

**Experimental setup.** In all our experiments reported below, we use publicly available model endpoints for inference [4, 23] with default parameter settings. All of our code and data are publicly available at [15] for future research.

### 4.1 Overall performance

We benchmark a range of frontier open-source and proprietary models using MMTU. Table 3 gives an overview of their performance, as well as their cost information. Notably, reasoning models such as GPT-5, o3 and Gemini-2.5-pro, significantly outperform chat-oriented models, with OpenAI GPT-5 achieving the best score at 69.6%, highlighting the challenging nature of MMTU. Among open-source reasoning models, DeepSeek R1 attains the best performance with a score of 57.9%, which while strong, reveals a potential gap between leading proprietary and open-source models.

Our analysis of traces generated by reasoning models reveals that they excel on MMTU over chat models, because of their strong coding skills, and abilities to break complex tasks on large tables, into a sequence of more manageable sub-tasks with smaller table context (i.e., subsets of rows and columns relevant to the task at hand).

From a cost-efficiency perspective[1], looking at the "cost per question" column in the table, several light-weight reasoning models (e.g., GPT-5-mini and Gemini-2.5-flash) are quite competitive in terms of both quality and cost (even after accounting for their intermediate thinking tokens), making them cost-effective choices for complex table tasks.

While we benchmark a broad range of models, our detailed analysis in the remainder of the paper will focus on four leading models to avoid clutter: two reasoning models (OpenAI GPT-5 and DeepSeek R1) and two chat-oriented models (OpenAI GPT-5-Chat and DeepSeek V3).

Figure 5 provides a detailed comparison across 10 task categories, between these four leading models. While reasoning models outperform chat-based models, the relative empty nature of the radar chart reveals substantial gaps in model performance, highlighting opportunities for improvement. For instance, we can see that models still face difficulties with table-centric coding tasks (e.g., Column

---

[1]Price-per-token figures are sourced from the respective API providers [27–34].

Transform, Table Transform, etc.). Tasks that require holistic reasoning across multiple tables and multiple columns (e.g., Table Join, Column Relationship, Data Cleaning, etc.) also remain challenging. We will present an error-analysis in Section 4.3.

A more granular breakdown across all 25 individual tasks is shown in Figure 4, where reasoning models (represented by shaded bars) noticeably outperform chat models on many complex tasks. Additional dataset-level performance results can be found in Table 5 in the appendix.

## 4.2 Detailed analysis and sensitivity

We highlight key analysis in this section, including long contexts, robustness to table perturbations, and sensitivity to format variations.

**Long table context.** Figure 6 shows the impact of long table context on model performance. In this analysis, we bucketize all questions within each task into four quartiles based on the token length of the associated tables. We then evaluate model accuracy within each quartile and aggregate the results across tasks, as shown in the figure. Across all four frontier models, performance consistently declines as the table context length increases (from left to right within each group).

These findings suggest that, despite recent advances in long-context LLMs [51, 36], long contexts remain a significant challenge for tables. In Appendix E, we use a detailed experimental comparison between the standard "needle-in-a-haystack (NIH)" [3, 110], and our table-based "needle-in-a-haystack in table (NIHT)" test – while frontier models perform nearly perfectly on NIH, their performance drops sharply on NIHT, revealing fundamental limitations in their ability to handle long table contexts.

**Robustness to table permutation.** Figure 7 illustrates model performance under different table permutations. Specifically, we randomly shuffle rows and/or columns in input tables[2], with the 4 bars in each group representing: (1) no shuffle, (2) row-only shuffle, (3) column-only shuffle, and (4) both row and column shuffle. Recall that unlike natural-language text, two dimension relational tables are permutation-invariant [94, 62], meaning that shuffling rows and columns should generally not change their semantic meanings and the associated tasks (e.g., the tasks in Figure 1 will remain the same, even if the rows/columns in the associated tables are shuffled).

However, the results show a consistent decline in model performance as we move from no permutation to row, column, and full (row + column) shuffling. Notably, column shuffling leads to a steeper decline than row shuffling. This suggests that despite their capabilities, language models that are pretrained primarily on linear, left-to-right text can remain sensitive to the structural ordering of tables, indicating a lack of robustness to alternate but semantically equivalent table layouts.

**Table format variations.** Figure 8 shows different models' performance using common table input formats: Markdown, CSV, JSON, and HTML. We observe that unlike prior studies that showed notable sensitivity of LLMs [119] to table formats, our results indicate that today's frontier models, especially reasoning ones (GPT-5 and R1), are becoming less sensitive to these format variations (except HTML that still lags behind other formats). This reduces the need for format-specific optimization as models continue to improve. For chat models (GPT-5-Chat and DeepSeek-V3), we note that using JSON has an advantage on MMTU, mainly because it is easier for models to identify value/column correspondence in the JSON format, especially in long table context settings (e.g., in needle-in-a-haystack tasks like NIHT and NIHI shown in Table 1).

## 4.3 Error analysis

We sampled 10 questions from each task, to manually analyze the underlying reason of these errors. We categorize all model errors into 4 main categories below:

**Table understanding (38%).** Table understanding is the largest category of errors in our analysis. Figure 9 shows a simple example from the Data Imputation task that is intuitive to understand. While the model correctly identifies the person's name for the missing cell, it filled in the value "`D. H. McFadden`" that is in an abbreviated form, inconsistent with other values in the same column that use the full-name format (the correct answer should be "`David Henry McFadden`").

---

[2] In shuffling columns, we keep the 3 left-most columns in all tables intact, as these are likely the "key columns" or "entity columns" in tables [47, 45] in tables, to best preserve the meaning of these tables.

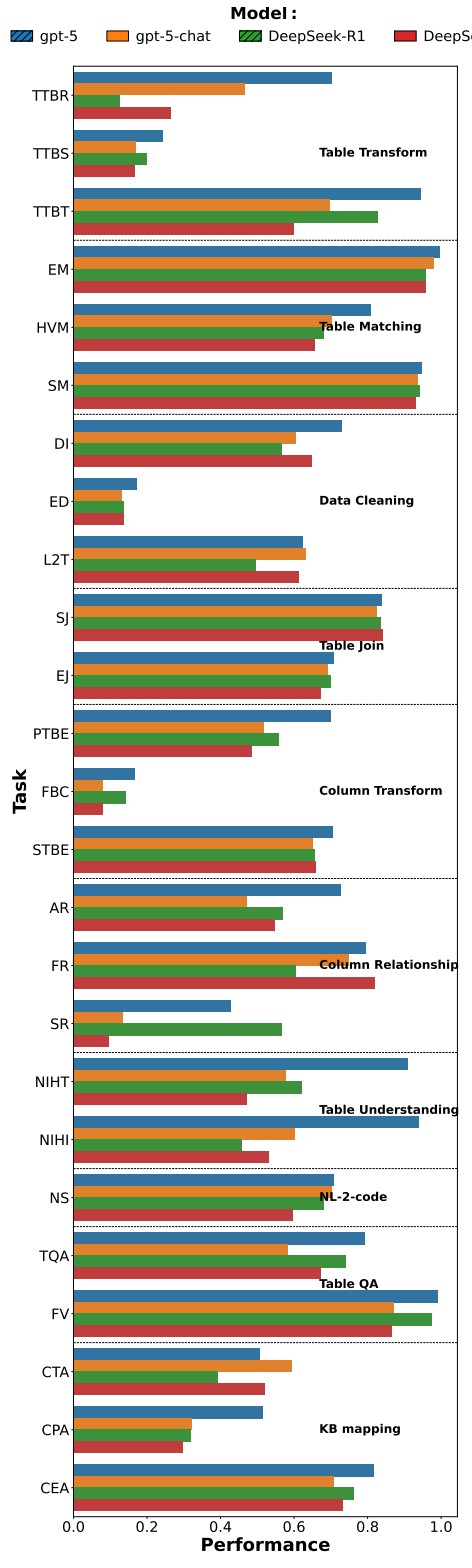

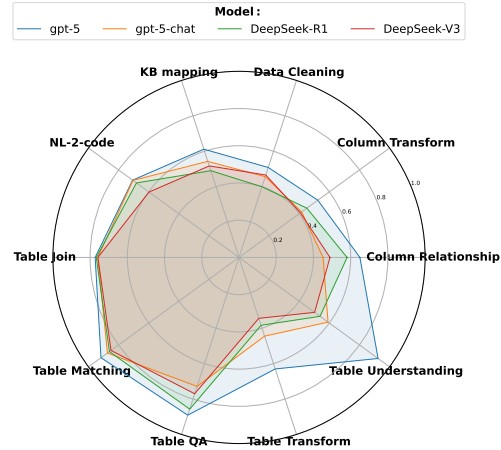

Figure 5: Performance comparison of frontier models in all 10 task-categories.

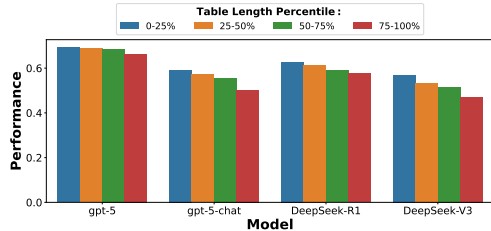

Figure 6: Impact of long table context on model performance.

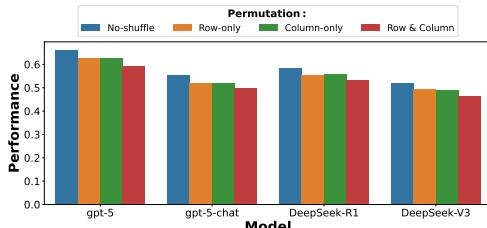

Figure 7: Impact of table-level permutation on model performance.

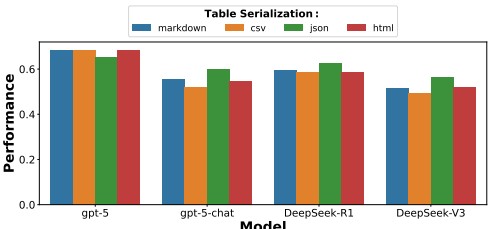

Figure 8: Impact of table format (markdown/csv/json/html) on model performance.

Figure 4: Performance comparison in all 25 tasks, for chat-models GPT-5-Chat and DeepSeek-V3 (solid bars), as well as reasoning models DeepSeek-R1 and OpenAI GPT-5 (shaded bars). On average, reasoning models are noticeably better.

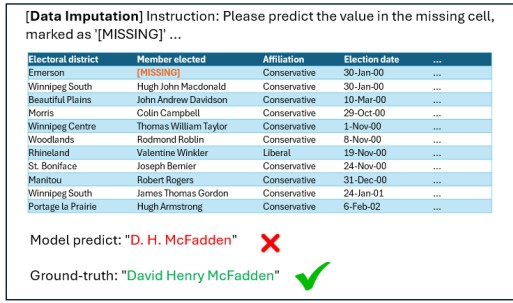

Figure 9: (Left): An example "table understanding" error in Data Imputation: while the model knows the person's name for the missing cell, it misses the table context and uses an abbreviated form of the name, instead of the full-name in the ground-truth. (Right): Another example "table understanding" error in Table Reshaping: while the model knows the correct transforms to perform ("explode" in Pandas), it miscounts the column index for the target column "Season(s)", which should be the sixth column instead of the fifth based on the reasoning trace, which leads to an incorrect transformation.

We observe that models are prone to errors when handling long table contexts, such as multiple tables or large tables with many rows and columns, as illustrated in Figure 6. Figure 9 (Right) shows a concrete example of the issue in the task of table-reshaping (TTBR in Table 1), where the reasoning trace shows that the model miscalculates the column index of the target column. This issue of long-context table is also notable in tasks like List-to-Table (L2T), Data Imputation (DI), Equi-Join (EJ), and NL-to-SQL (NS), etc.

A more detailed analysis of challenges in long-context tables is provided in Appendix E, where we examine a table-specific variant of the "needle-in-a-haystack" task, to understand why long-context tables remain challenging for models.

**Reasoning and coding (28%).** We find models can often make mistakes on tasks that require coding and reasoning on top of tables. For instance, for tasks in Table Transform, Column Transform, and NL-2-Code categories, models can generate code (SQL or Pandas) that is "close" to the correct answer, but misses important details that require reasoning over table context holistically (e.g., data format across all cells in the same column), which leads to incorrect results.

**Knowledge (18%).** For tasks in the categories of KB mapping (CEA and CTA), Semantic Join (SJ), and Data Imputation (DI), we find models can sometimes hallucinate facts (e.g., in the case of CEA, creating knowledge base references that do not exist), or recite inaccurate facts (e.g., in the case of Semantic Join and Data Imputation).

**Other (15%).** The remaining issues, such as result extraction, timeout and context limitation in response generation, and possible ambiguity in questions/ground-truth, fall in this category.

LLM-based error analysis. In addition, we complement the manual error analysis, by instructing LLM to inspect the question and ground-truth, to classify model prediction errors into these categories. We present the corresponding results in Appendix F in the interest of space.

## 5 Conclusion

In this work, we present MMTU, a comprehensive benchmark designed to assess foundation models on a broad range of real-world table tasks. By focusing on real-world tasks that professional data engineers, data analysts, and database administrators often have to face, MMTU poses expert-level challenges for foundation models in table understanding and reasoning.

Future work includes broadening the scope of table tasks to cover those that are important in practice but underrepresented in the existing research literature, as well as incorporating more subjective or creative tasks such as data generation, summarization, and enrichment. Another promising direction is the integration of multi-modal models for evaluation, which may have an advantage over text-only models on tasks that require two-dimensional table understanding.

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

# A  Broader Discussions

## A.1  Open access to data and code

Our benchmark data is available at `https://huggingface.co/MMTU-benchmark`, and our evaluation code is available at `https://github.com/MMTU-Benchmark`.

## A.2  License for existing assets

We provide comprehensive references to the sources of our data assets throughout this paper, with more details in [15]. We respect and conform to the licensing terms of existing datasets, whenever we can find such terms when we create our benchmark.

# B  Task overview

## B.1  Column transformation

Column transformation refers to the category of tasks, where new columns in a table are derived using existing columns. We select three key tasks in this category:

**Program Transform by Example (PTBE)** [76, 70, 67, 116, 77]. This is a popular task studied in the data management and programming language literature. In this task, we are given a few example output values in a new output column (usually provided by users as demonstrations), and the task is to synthesize a transformation program using input tables, such that the produced output can match the user-given output examples. Figure 1 shows an example of this task. We use the benchmark datasets from [76, 70] for this task in MMTU.

**Semantic Transform by example (STBE)** [75, 35, 64]. This task is similar to PFBE above, in that users also provide a few example output to demonstrate the intended transformations, but the target transformation requires "semantic" transformations (e.g., country to capital-city, or company to stock-ticker), that cannot be programmed using a syntactic program like in PFBE. We use the datasets from [75, 35]for our benchmark.

**Formula by Context (FBC)** [52, 54]. This task is studied in the context of spreadsheets, where given a target spreadsheet cell in which users want to enter a spreadsheet formula, we are asked to predict the intended formula in the target cell. We use the 4 benchmark datasets in [52].

## B.2  Table Transformation

In the table transformation task category, we also need to perform transformations, but unlike "Column transformations" above where only new columns are derived (without changing the shape of the input table), in "Table transformation", a broader class of transformations can be invoked (e.g., table reshaping, restructuring, group-by, aggregation, etc.), which can change the form and shape of the original input table.

**Table Transform by Output Table (TTBT)** [40, 125, 122, 142]. In this task, we are given a pair of input/output tables, and the task is to infer a transformation program (in SQL or Pandas), which when executed on the input table, can generate the desired output table. Figure 1 shows an example of this task. We use the datasets from [40, 125].

**Table Transform by Output Schema (TTBS)** [135, 113, 114]. This task is similar to TTBT above, except that the output table is a schematic depiction of how the desired output table should look like, without being the exact output that the desired transformation program would generate on the given input (as it is sometimes hard to generate such output table without first producing the desired program). We use the dataset in [135]for this task.

**Table Transform by Relationalization (TTBR)** [93, 39, 83, 132, 87]. Because relational data analysis often requires input tables to be in a relational form, this task transforms data in non-relational semi-structured forms, into the standard relational form. The task is to predict such a relationalization transformation program, based on the characteristics of the input table. We use the dataset in [93].

### B.3 Table matching

In this task category, we try to match relevant rows and columns between multiple tables.

**Entity Matching (EM)** [65, 103, 146, 95, 105]. Entity matching is the task of determining whether rows or entities from two tables correspond to the same real-world entity. We use the datasets in [65, 103]for benchmarking.

**Schema Matching (SM)** [86, 145, 41, 100]. Schema matching tries to match relevant columns from two tables that map to the same semantic concept. Figure 1 shows an example of this task. We use datasets in [86, 145]for MMTU.

**Header Value Matching (HVM)** [94, 108]. In HVM, we are given a table without headers, and a shuffled list of the original headers in the table, where the task is to match column headers with column values in tables. We use the dataset in [94]for benchmarking.

### B.4 Data cleaning

Tasks in the data cleaning category tries to improve the quality of input tables, which are popular tasks studied in the data management literature.

**Data Imputation (DI)** [45, 68, 94]. Data imputation is the intuitive task of predicting missing values in a relational table, based on the surrounding context of the table. Figure 1 shows an example of this task. We use the dataset from [45, 68].

**Error Detection (ED)** [60, 126, 101, 50]. The task of Error detection aims to identify erroneous values in a table that are semantically inconsistent or anomalous with the rest of the column. We use the dataset in [50].

**List to Tables (L2T)** [59, 46, 69, 90]. In List-to-table, the task is to segment records of data without clear separators, into columns, so that values in the same column are homogeneous, and the resulting table becomes a natural relational table, with values consistently aligned in homogenous columns. We use the dataset from [59].

### B.5 Table join

Join is an important operation that connects multiple related tables together.

**Equal Join (EJ)** [96, 56, 111, 81]. In Equal-join, we are given a collection of related tables, and the task is to identify all join relationships between the tables. Figure 1 shows an example of this task. We use the dataset in [96].

**Semantic Join (SJ)** [75, 35, 64, 128]. In Semantic Join, we are also tasked to join related tables together, but instead of the common equi-join, which is based on string-equality comparisons, the join relationship is based on semantic relatedness (e.g., country and capital city, or company and stock ticket). We use the same semantic-transformation datasets from [75, 35], but converting the input/output transformation columns, as join keys from two separate tables.

### B.6 Column relationships

In this category of tasks, the goal is to identify implicit but semantically meaningful relationships from an input table.

**Arithmetic Relationship (AR)** [115, 1]. This task focuses on identifying arithmetic relationships from an underlying table. Figure 1 shows an example of this task.

**String Relationship (SR)** [1, 76, 71]. This task focuses on identifying string transformation relationships from a given table.

**Functional Relationship (FR)** [1, 106, 129]. This task focuses on identifying functional-relationships from input tables. We use the datasets from [1] our as benchmarks for all three tasks.

### B.7 Table understanding

Tasks in this category are intended to test models ability to understand tables, and retrieve relevant facts from tables.

**Needle-in-a-haystack-table (NIHT)**. In this task, we create a variant of the popular "Needle-in-a-haystack" task in the context of tables, like described in detail in Appendix E. We use the dataset from [68] to construct tests in this task.

**Needle-in-a-haystack-index (NIHI)**. In this task, we reverse the NIHT task above, and ask models to identify index positions corresponding to a given value in a table. We use the same dataset from [68] to construct tests for this task.

### B.8 NL-2-Code

**NL-2-SQL (NS)** [147, 92, 89, 138, 148]. NL-2-SQL is a popular task to translate natural-language questions into SQL statements that can execute given an input table. We use the benchmarks from [147, 92, 89, 138, 148] in MMTU.

### B.9 Table Question Answering (Table QA)

**Table-QA (TQA)** [131, 137, 57, 58]. Table QA is another popular task, where models are used to directly answer questions on a given input table, without code execution. We use benchmarks from [53, 131, 137, 57] for this task.

**Fact verification (FV)** [53, 136, 141]. Table fact verification is a variant of TQA, in which models are asked if a statement is refuted or supported by facts presented in an input table. We use data from [53] for this task.

### B.10 Knowledge-base mapping (KB Mapping)

KB mapping is the task where we map facts and relationships of a table, to known KB ontologies.

**Column type annotation (CTA)** [82, 134, 118, 140, 80]. This is a popular task where columns in a table is mapped to known entity types in a knowledge-base. We use the CTA dataset from [82].

**Column property annotation (CPA)** [82, 63, 104]. [82]. The CPA task is to predict the relationship of a given pair of columns, based on known properties in a knowledge-base. We use the CPA dataset from [82].

**Cell entity annotation (CEA)** [82, 66, 42, 130, 98]. The CEA task predicts the knowledge-base entity id of a given cell in a table. We use the CEA dataset from [82].

## C   Additional Statistics

Table 4 lists the exact ranges of token count calculated for MMTU questions, in each quartile and for each task.

## D   Detailed model performance at the dataset level

Table 5 shows a detailed breakdown of performance results on MMTU, at the dataset level. Reasoning models indeed outperform chat models in many cases, though chat models are better on knowledge-centric tasks like CTA, where we find reasoning models have a tendency to hallucinate (e.g., type names that do not exist in knowledge bases).

| Task | 0–25% range | 25–50% range | 50–75% range | 75–100% range |
|---|---|---|---|---|
| Arithmetic-Relationship | 1150.0–3547.0 | 3547.0–6834.0 | 6834.0–15601.5 | 15601.5–126654.0 |
| Cell-entity-annotation | 263.0–2138.5 | 2138.5–3706.0 | 3706.0–4693.5 | 4693.5–7526.0 |
| Column-type-annotation | 274.0–576.8 | 576.8–1346.5 | 1346.5–3094.8 | 3094.8–6744.0 |
| Columns-property-anotation | 261.0–612.8 | 612.8–1416.0 | 1416.0–3004.2 | 3004.2–6754.0 |
| Data-Imputation | 214.0–628.2 | 628.2–1320.0 | 1320.0–1971.0 | 1971.0–102929.0 |
| Data-transform-pbe | 262.0–309.8 | 309.8–333.5 | 333.5–412.2 | 412.2–21846.0 |
| Data-transform-reshape | 534.0–916.5 | 916.5–1692.0 | 1692.0–3621.0 | 3621.0–76096.0 |
| Entity-Matching | 175.0–207.0 | 207.0–232.0 | 232.0–248.5 | 248.5–425.0 |
| Error-Detect | 211.0–235.0 | 235.0–293.0 | 293.0–638.5 | 638.5–21040.0 |
| Formula-prediction-context | 269.0–1096.2 | 1096.2–1315.0 | 1315.0–1564.0 | 1564.0–10332.0 |
| Functional-Dependency | 1341.0–3641.0 | 3641.0–5157.0 | 5157.0–7573.0 | 7573.0–76083.0 |
| List-to-table | 184.0–283.5 | 283.5–352.0 | 352.0–514.5 | 514.5–2955.0 |
| NL2SQL | 418.0–872.0 | 872.0–1245.0 | 1245.0–3293.0 | 3293.0–35250.0 |
| Schema-Matching | 547.0–1803.5 | 1803.5–2378.0 | 2378.0–3737.5 | 3737.5–10562.0 |
| String-Relationship | 2961.0–7596.2 | 7596.2–14127.5 | 14127.5–26032.5 | 26032.5–92595.0 |
| Table-Fact-Verification | 269.0–451.0 | 451.0–584.0 | 584.0–836.5 | 836.5–2959.0 |
| Table-Locate-by-Row-Col | 11190.0–21158.0 | 21158.0–22433.0 | 22433.0–23731.0 | 23731.0–72074.0 |
| Table-QA | 229.0–381.0 | 381.0–537.0 | 537.0–847.2 | 847.2–13752.0 |
| Table-needle-in-a-haystack | 11123.0–21286.0 | 21286.0–22737.0 | 22737.0–23664.0 | 23664.0–72009.0 |
| Transform-by-input-output-table | 200.0–259.0 | 259.0–297.0 | 297.0–366.0 | 366.0–728.0 |
| Transform-by-output-target-schema | 360.0–1095.5 | 1095.5–1827.5 | 1827.5–2845.8 | 2845.8–113819.0 |
| equi-join-detect | 477.0–2471.0 | 2471.0–3972.0 | 3972.0–6741.0 | 6741.0–107690.0 |
| header-value-matching | 186.0–260.0 | 260.0–338.5 | 338.5–489.2 | 489.2–1102.0 |
| semantic-join | 243.0–367.5 | 367.5–742.0 | 742.0–1860.5 | 1860.5–19184.0 |
| semantic-transform | 209.0–247.0 | 247.0–265.0 | 265.0–285.0 | 285.0–454.0 |

Table 4: Exact ranges of token count calculated for MMTU questions, in each quartile and for each task, as used in long-table-context analysis (Figure 6). The number of tokens is calculated using GPT-4o tokenizer.

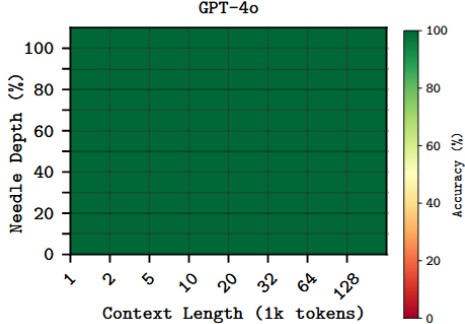

Figure 10: The needle-in-a-haystack test used in NLP context. Each cell in the heatmap corresponds to a test to retrieve a simple fact planted in a document, for a given document length and retrieval depth. Frontier models like GPT-4o and Gemini-2.5 can now typically achieve perfect accuracy, as shown by the all-green heatmap (results from [110]).

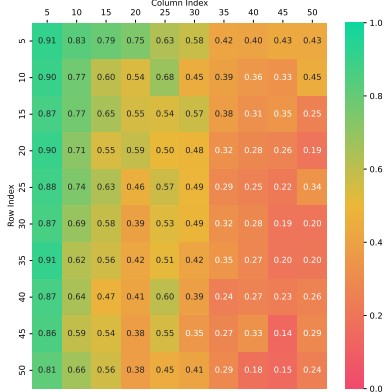

Figure 11: Our table-based needle-in-a-haystack test, where frontier models like GPT-4o continue to struggle to retrieve simple facts from large tables, especially with more columns (notice the asymmetry in the heatmap, where an increased column-index makes the task significantly harder).

# E Long-context table understanding: A case study of "needle in a haystack in tables"

In this section, we take a closer look at one simple task in MMTU that we refer to as "needle in a haystack in tables", to more deeply analyze the problem of table understanding with long table context (large tables with many rows and columns).

Recall that "Needle-in-a-haystack (NIH)" [3, 110], is a popular test traditionally used to evaluate long-context LLM's ability to retrieve a simple fact (a needle) from a long document context (e.g., 100K or 1M tokens). Recent advances in frontier long-context LLMs have made this task almost irrelevant, as most frontier models such as GPT-4o, Llama-4, and Gemini-2.5 can now score perfectly on NIH [97, 120, 26, 14], like the heatmap in Figure 10 would show. Here the all-green heatmap

| Task Name | Dataset | Chat Model | | Reasoning Model | |
|---|---|---|---|---|---|
| | | GPT-5-Chat | DeepSeek-V3 | GPT-5 | Deepseek-R1 |
| Data-transform-reshape | Auto-Tables | 0.500 | 0.282 | **0.710** | 0.126 |
| Transform-by-output-target-schema | commercial-pipelines | 0.231 | 0.154 | **0.385** | 0.154 |
| | github-pipelines | 0.253 | 0.224 | **0.269** | 0.213 |
| | **Average** | 0.242 | 0.189 | **0.327** | 0.183 |
| Transform-by-input-output-table | AutoPandas | 0.593 | 0.481 | **0.889** | 0.852 |
| | Scythe | 0.667 | 0.600 | **0.950** | 0.883 |
| | **Average** | 0.630 | 0.541 | **0.919** | 0.868 |
| Entity-Matching | Amazon-Google | 0.872 | 0.856 | **0.938** | 0.868 |
| | BeerAdvo-RateBeer | 0.966 | 0.955 | **0.989** | 0.943 |
| | DBLP-ACM | 0.983 | 0.985 | **0.997** | 0.973 |
| | DBLP-Scholar | 0.962 | 0.962 | **0.976** | 0.950 |
| | Fodors-Zagats | **1.000** | 0.995 | **1.000** | 0.989 |
| | Walmart-Amazon | 0.948 | 0.940 | **0.983** | 0.940 |
| | iTunes-Amazon | 0.953 | 0.943 | **0.991** | 0.915 |
| | **Average** | 0.955 | 0.948 | **0.982** | 0.940 |
| header-value-matching | TableGPT | 0.699 | 0.633 | **0.818** | 0.678 |
| Schema-Matching | DeepMDatasets | **1.000** | **1.000** | **1.000** | **1.000** |
| | HXD | 0.928 | **0.945** | 0.937 | 0.937 |
| | Wikidata | 0.897 | 0.882 | 0.912 | **0.931** |
| | assays | 0.843 | 0.913 | 0.887 | **0.924** |
| | miller2 | 0.916 | **0.946** | 0.925 | 0.929 |
| | prospect | 0.976 | 0.927 | **0.996** | 0.958 |
| | **Average** | 0.927 | 0.935 | 0.943 | **0.947** |
| Data-Imputation | WebTable | 0.493 | 0.483 | **0.642** | 0.499 |
| | tablib | 0.704 | 0.627 | **0.843** | 0.664 |
| | **Average** | 0.599 | 0.555 | **0.742** | 0.582 |
| Error-Detect | Relational-Tables | 0.136 | **0.214** | 0.211 | 0.153 |
| | Spreadsheet-Tables | 0.103 | 0.123 | **0.128** | 0.113 |
| | **Average** | 0.119 | 0.168 | **0.170** | 0.133 |
| List-to-table | TEGRA | **0.604** | 0.600 | 0.598 | 0.567 |
| semantic-join | DataXFormer | 0.797 | 0.759 | 0.834 | **0.840** |
| | SEMA-join | 0.899 | 0.889 | **0.947** | 0.928 |
| | **Average** | 0.848 | 0.824 | **0.890** | 0.884 |
| equi-join-detect | Auto-BI | 0.665 | 0.655 | **0.692** | 0.688 |
| Data-transform-pbe | TDE | 0.487 | 0.419 | **0.708** | 0.614 |
| | Transformation-text | 0.515 | 0.479 | **0.681** | 0.636 |
| | **Average** | 0.501 | 0.449 | **0.694** | 0.625 |
| Formula-prediction-context | cisco-random | 0.124 | 0.101 | **0.224** | 0.166 |
| | enron-random | 0.051 | 0.043 | **0.163** | 0.105 |
| | pge-random | 0.071 | 0.080 | **0.116** | 0.125 |
| | ti-random | 0.076 | 0.051 | **0.179** | 0.140 |
| | **Average** | 0.081 | 0.069 | **0.170** | 0.134 |
| semantic-transform | DataXFormer | 0.423 | 0.419 | **0.507** | 0.443 |
| | SEMA-join | 0.877 | 0.865 | **0.915** | 0.883 |
| | **Average** | 0.650 | 0.642 | **0.711** | 0.663 |
| Arithmetic-Relationship | Auto-Relate | 0.476 | 0.506 | **0.735** | 0.693 |
| Functional-Dependency | Auto-Relate | 0.739 | 0.730 | **0.799** | 0.762 |
| String-Relationship | Auto-Relate | 0.148 | 0.128 | 0.388 | **0.551** |
| Table-needle-in-a-haystack | MMTU | 0.546 | 0.446 | **0.920** | 0.703 |
| Table-Locate-by-Row-Col | MMTU | 0.570 | 0.517 | **0.941** | 0.710 |
| NL2SQL | Archer | 0.327 | 0.183 | 0.394 | **0.404** |
| | KaggleDBQA | **0.535** | 0.465 | 0.524 | 0.481 |
| | Spider | **0.785** | 0.776 | 0.739 | 0.707 |
| | WikiSQL | **0.745** | 0.733 | 0.732 | 0.704 |
| | bird | 0.463 | 0.429 | **0.478** | 0.459 |
| | **Average** | 0.571 | 0.517 | **0.574** | 0.551 |
| Table-QA | FinQA | 0.211 | 0.229 | **0.497** | 0.403 |
| | TableBench | 0.441 | 0.432 | **0.580** | 0.566 |
| | WikiQA | 0.728 | 0.685 | **0.827** | 0.797 |
| | **Average** | 0.460 | 0.449 | **0.635** | 0.589 |
| Table-Fact-Verification | TabFact | 0.852 | 0.838 | **0.940** | 0.927 |
| Column-type-annotation | SemTab2019 | **0.554** | 0.409 | 0.462 | 0.371 |
| Columns-property-anotation | SemTab2019 | 0.296 | 0.258 | **0.492** | 0.280 |
| Cell-entity-annotation | SemTab2019 | 0.687 | 0.676 | **0.803** | 0.689 |

Table 5: Dataset-level Performance of Frontier Chat & Reasoning Models

indicates that this particular LLM under test, GPT-4o, can perfectly retrieve a needle planted at varying depth (y-axis), within documents of varying lengths (x-axis), with 100% accuracy.

| Super-Categories | Percentage |
|---|---|
| Reasoning & Coding Errors | 40.3% |
| Table-Understanding Errors | 25.6% |
| Knowledge Errors | 16.4% |
| Other Errors | 11.7% |

Table 6: Distribution of Super-category Errors On Incorrect Questions by o4-mini

| Super Categories | Sub Categories | Percentage |
|---|---|---|
| Table-Understanding Errors | Row/Column index misalignment | 0.283122 |
| Reasoning & Coding Errors | Incorrect reasoning | 0.223264 |
| Reasoning & Coding Errors | Semantically incorrect code | 0.138167 |
| Knowledge Errors | Incorrect knowledge | 0.105523 |
| Knowledge Errors | Hallucinated facts | 0.0334465 |
| Other Errors | Format error / Result-extraction failure | 0.0323357 |
| Other Errors | Ground-truth ambiguity / quality | 0.0323357 |
| Table-Understanding Errors | Complex table | 0.0156742 |
| Table-Understanding Errors | Multi-table understanding | 0.0153039 |
| Knowledge Errors | Numerical Errors | 0.014008 |

Table 7: Distribution of Top-10 Sub-category Errors On Incorrect Questions by o4-mini.

We adapt NIH to the table setting, and study the table-version of the "needle in a haystack", which we call "needle-in-a-haystack-in-table" (NIHT), that tests LLM's ability to retrieve a simple fact (needle) within a cell of a table. Specifically, in NIHT, we randomly sample 100 real tables with at least 50 rows and columns, and ask LLMs to retrieve a simple fact randomly planted at row-i and column-j of each table, repeated over all (i, j) positions. LLM responses at different (i, j) positions can then be compared with the ground-truth (the cell value at those positions) to measure accuracy.

We would like to highlight that NIHT is the simplest possible task, for table understanding in long table context, as it performs a simple retrieval with no additional steps. Any real table tasks with long table context (e.g., Table QA on large tables) will necessarily be at least as hard as NIHT, as those tasks will need to first retrieve/identify relevant cells in long table context, before further processing (reasoning, calculation or coding) can be performed, making NIHT a good case study for long-context table understanding.

The results of NIHT are illustrated also using a heatmap in Figure 11. As we can see, compared to Figure 10, today's frontier LLMs still face substantial challenges in correctly identifying a cell value within a two-dimensional table, like indicated by the red cells in the heatmap. This is especially true when we increase the number of columns (e.g., with more than 25 columns, LLM accuracy quickly falls below 0.5). Furthermore, we observe a *strong asymmetry* in the heatmap – e.g., with 50 rows and 5 columns we still have a reasonable accuracy of 0.81 (at the lower-left corner of the heatmap), but with 5 rows and 50 columns the accuracy drops to a lowly 0.43 (at the top-right corner), underscoring the challenge LLMs face when reading "vertically" in the column direction, which are consistent with our intuition that LLMs pre-trained on one-dimensional natural-language texts are less effective in reading two-dimensional tables "vertically", like was also observed in [94, 119].

# F   LLM-based Error analysis

**Setup.** We conducted a LLM-based error analysis to understand the underlying reasons of these errors on MMTU. We built a detailed categorization of typical errors that we observe in models' answers, using the following 4 super-categories, and 14 sub-categories:

- **Super-category: Table-Understanding Errors**
  - Row/Column index misalignment (e.g., the row/column index is not correctly recognized in a table, leading to incorrect model output)
  - Long context (e.g., long table causes model to produce incorrect results)

- Multi-table understanding (e.g., join relationships between multiple tables are identified incorrectly, leading to incorrect answers)
- Complex table (e.g., merged cells spanning multi-row/cols, hierarchical headers, etc., causing models to produce incorrect answers)

- **Super-category: Reasoning & Coding Errors**
  - Syntax or execution error (if the task asks for SQL/Python code, and the code should encounter an execution error)
  - Semantically incorrect code (if the task asks for SQL/Python code, and the code is executable but the result is incorrect)
  - Incorrect reasoning (pick this category only if the model's reasoning shows obvious flaws that directly lead to incorrect answers)

- **Super-category: Knowledge Errors**
  - Hallucinated facts (e.g., in CEA, output knowledge base references that do not exist)
  - Incorrect knowledge (e.g., in data imputation, filled in a related but incorrect value)
  - Numerical Errors (e.g., in data imputation, filled a numerical value that is close but not precise)

- **Super-category: Other Errors**
  - Format error / Result-extraction failure
  - Timeout
  - Ground-truth ambiguity / quality
  - None of the above: create a new category, and give a short description

**Result.** We used o4-mini to categorize all 11K questions for which o4-mini were incorrect on MMTU into these categories. The distribution of super-categories is shown in Table 6, note that the percentage does not sum to one due to format issue and LLM curated super-categories. The distribution of sub-categories is shown in Table 7. We observe that reasoning & coding, and table understanding remain the two largest categories of errors.

