# OpenReview forum: "MMTU: A Massive Multi-Task Table Understanding and Reasoning Benchmark"
_NeurIPS.cc/2025/Datasets_and_Benchmarks_Track — NeurIPS 2025 Datasets and Benchmarks Track poster_

### Official Review · Reviewer_BG3j · 2025-06-24

**Rating:** 5
**Confidence:** 4

**Summary:**

This paper presents MMTU (Massive Multi-Task Table Understanding), a comprehensive benchmark designed to evaluate the capabilities of foundation models on structured data. MMTU includes over 30,000 questions spanning 25 diverse, real-world table tasks, selected through a literature review covering data management, programming languages, and web-based table use cases. The authors standardize task formats into a unified triple-based formulation and curate the data to ensure consistency. They further introduce a common evaluation framework supporting multiple metrics and task types. MMTU enables rigorous analysis of model behavior under long-context tables, table row/column permutations, and format variations, and provides an initial error categorization to guide future research in table understanding.

**Additional Feedback:**

Page 8 could be more efficiently organized by converting Figures 4, 6, 7, and 8 into tables, which would reduce visual clutter and save space for additional content.

**Dataset Code Accessibility:**

Yes

**Dataset Code Comments:**

The dataset can be easily accessed on Hugging Face and the github repository is public.

**Ethical Considerations:**

No, there are no or only very minor ethics concerns

**Final Justification:**

The authors of the paper satisfactorily addressed all of my concerns. However, I still believe that the pipeline should eliminate the human verification step, as the majority of verification (over 99%) is conducted using LLM-As-Judge after the rebuttal. The authors should also include all the details discussed in this rebuttal, as most new insights and clarifications come from it. In summary, I believe the literature will greatly benefit from this dataset.

**Limitations Weaknesses:**

While the paper makes strong contributions, several limitations remain. The methodology behind dataset selection and standardization lacks full transparency, particularly regarding the use of DeepResearch and the rationale for filtering based on privacy or security. The evaluation framework, though valuable, is currently hard to extend with new metrics and lacks formal definitions for key components. Experimentally, the comparison across models is limited in scope and fairness—only a few models are evaluated, with imbalanced parameter sizes, and important baselines like Gemini and TableGPT are missing. Finally, the error analysis is underdeveloped and would benefit from more explicit task-specific definitions and a more systematic categorization of failure modes.
Here are more details:
1. The systematic review using DeepResearch-like tools should be better contextualized, particularly by specifying the prompts used. This would enhance both reproducibility and the potential for future follow-up studies.
2. It remains unclear to what extent the review relies on keyword-based searches over paper titles, and what role the DeepResearch tool actually plays. Clarifying this division would help in assessing the rigor of the review process.
3. Regarding task selection, you state that the tasks are "objectively evaluable." However, I wonder if this claim is completely true. In domains like Text-to-SQL, recent work has shown that tasks often suffer from ambiguity due to the ambiguous nature of natural language [1]. These concerns might also apply to the collection of other datasets. I recommend applying the same LLM-based shallow filtering that you already use for tasks involving privacy or security concerns (as mentioned around line 158) to address this issue.
4. The rationale for filtering datasets based on privacy or security is not clearly stated in the paper. Why would the collected datasets raise such concerns? This needs to be justified and made explicit.
5. The paper does not provide detailed statistics on each dataset's contributions or clarify the level of data curation applied. Have the datasets been manually checked for errors or inconsistencies? For example, BIRD is known to have limitations [2], and since it is included via MMTU, these issues should be acknowledged.
6. I appreciate the effort to provide a common evaluation framework. However, its extensibility is unclear. For instance, how challenging is it to introduce new metrics, such as the one in [3] for Text-To-SQL, into the benchmark? Based on the current version of the codebase at the time of this review, there does not seem to be support or documentation for easily adding new metrics.
7. During the review, I noticed that the code implementation for Execution Accuracy seems to penalize column order, as the tuples are not sorted before checking their equality [4]. This penalizes predicted queries when the projected columns are correct but swapped. While this may not significantly affect the results, I recommend rechecking the code and updating the results if necessary. However, since I have not thoroughly examined the code, I am merely raising a concern for the authors to address, and I will not factor this point into my review score. If this issue is confirmed, I suggest updating the results.
8. Regarding Python code generation, what safety measures are implemented when executing model-generated code? From the paper, it seems that no sandboxed environment is used. I was expecting something like E2B [5] or similar secure execution solutions. This is a significant omission and should be addressed.
9. The evaluation framework would benefit from additional formalism: Line 167 mentions a "structured yet open-ended answer format"—this should be mathematically defined, as the structure of the expected answer can significantly impact model performance. The metrics used for tasks not yet employed to evaluate foundation models (line 149) should also be explicitly defined.
10. The "Expert verification" step (Figure 2) is based on only 20 samples per task, which is insufficient to draw reliable conclusions. I recommend either removing the step from the figure or significantly expanding the verification. Furthermore, if all datasets are sourced from human-annotated corpora, the motivation for additional verification is unclear. What specific challenges of your standardization process warrant further human validation?
11. Given the stochasticity of reasoning models—especially at high temperatures—I strongly recommend executing each model multiple times and averaging the results, to ensure robustness.
12. The evaluation should also include Gemini models. Recent studies [6] have shown that Gemini does not suffer the same long-context limitations affecting other LLMs. Another significant omission is the absence of TableGPT, which has demonstrated strong performance in table reasoning and should be included for a more comprehensive comparison.
13. The experimental setup currently includes only four models, limiting the findings' generalizability. Moreover, comparing DeepSeek-R1 (~600B parameters) against LLaMA 70B is not a fair comparison. This setup does not allow the conclusion that "reasoning models outperform chat models" to be made reliably. You should consider adding larger chat models (e.g., Gemini 1.5 Pro, LLaMA 405B) and more reasoning models (e.g., QwQ 32B, Qwen 3 family) to balance the evaluation.
14. The error analysis section is appreciated, but currently too vague. For example, the term “table understanding” is used broadly without being clearly defined across the diverse tasks in MMTU. Furthermore, it is unclear whether the 10 analyzed samples were drawn from chat or reasoning model failures—this distinction could significantly change the interpretation. I suggest introducing formal error categories and using them consistently across tasks to communicate the findings better.

[1] Saparina, Irina, and Mirella Lapata. "Ambrosia: A benchmark for parsing ambiguous questions into database queries." Advances in Neural Information Processing Systems 37 (2024): 90600-90628.

[2] Wretblad, Niklas, et al. "Understanding the effects of noise in text-to-SQL: an examination of the BIRD-bench benchmark." arXiv preprint arXiv:2402.12243 (2024).

[3] Papicchio, Simone, Paolo Papotti, and Luca Cagliero. "Qatch: Benchmarking sql-centric tasks with table representation learning models on your data." Advances in Neural Information Processing Systems 36 (2023): 30898-30917.

[4] https://github.com/MMTU-Benchmark/MMTU/blob/04890ae184fe9773a8b7297902c7ea4818c0a5ac/evaluators/nl2sql.py#L23

[5] https://e2b.dev/docs

[6] Chung, Yeounoh, et al. "Is Long Context All You Need? Leveraging LLM's Extended Context for NL2SQL." arXiv preprint
arXiv:2501.12372 (2025).

**Strengths Contributions:**

I believe in this paper's contribution. Due to the current limitations (see later), I will initially set 2 as a reject, but I am willing to change it during the rebuttal.

- The work is timely and relevant, given the surge of interest in applying foundation models to structured data and the ongoing push for more robust, task-agnostic benchmarks.
- The paper is clearly written and well-structured. Most questions that arise during reading are thoughtfully addressed within the text, which reflects strong attention to clarity and organization.
- Introducing a large-scale benchmark for table understanding is a highly valuable contribution. It addresses a real need in the literature for a unified and comprehensive evaluation across diverse tasks.
- The authors' effort in standardizing 25 heterogeneous tasks into a unified, simple, yet expressive triple-based problem formulation is impressive. This level of abstraction provides both generality and usability, enabling meaningful comparisons across task types.
- Despite its current limitations (see limitations), developing a unified evaluation framework lays valuable groundwork for standardized comparisons across models and tasks.

---

> ### Author Rebuttal · Authors · 2025-07-31
>
> We thank the reviewer for your kind encouragement. We have carefully studied the limitations raised in the review, which we address as follows:
>
> ***Additional details on the use of DeepResearch-like tools for systematic reviews of relevant research literature***
>
> - Thank you for the suggestion. We will add details in the revised manuscript about how we employed DeepResearch-like tools to complement traditional keyword-based search methods, which will enhance the reproducibility of our literature review and facilitate future follow-up work.
>
>
> ***Whether certain tasks like Text-to-SQL are entirely objectively evaluable, and how to extend the existing benchmarking framework to additional evaluation metric, e.g., for Text-to-SQL***
>
> - While Text-to-SQL tasks are typically evaluated using objective metrics such as Execution Match, which is a tradition that we follow, we acknowledge that certain edge cases may benefit from alternative evaluation criteria. We emphasize that our benchmarking framework is devised to be extensible by design. In practice, integrating new evaluation logic (e.g., supporting alternative correctness criteria) is straightforward and involves replacing a modular “evaluator” function within ~10s of lines of code. We have documented this extensibility in our GitHub repository to support community reuse and extension.
>
>
> ***Potential errors or inconsistency in datasets like BIRD, where the ground-truth labels may not be perfect***
>
> - We recognize that some datasets, including BIRD, may contain imperfect ground-truth annotations. While we conducted manual checks on sampled questions across datasets to identify and mitigate systematic issues (e.g., in formatting and evaluation), we acknowledge imperfect ground-truth we inherit as a limitation. In response, we will include a dedicated section in the revised manuscript summarizing known labeling issues to the best of our knowledge in the datasets we use, in order to best inform future research.
>
> ***The rationale for filtering datasets based on privacy or security***
>
> - While we have not identified any clear privacy or security risks in the data, we nevertheless apply the filtering steps in accordance with our organizational policies, and to mitigate potential issues that may escape human review.
>
> ***Python code generation and safety measures***
>
> - We would like to clarify that all generated code can indeed be executed in a sandboxed container environment to ensure safety and prevent unintended effects. Detailed instructions for containerized evaluation are now added in our repository’s README. We thank the reviewer for highlighting the importance of this clarification.
>
>
> ***Clarification on “"structured yet open-ended answer format"”***
>
> - We appreciate the feedback on terminology and agree that “structured yet open-ended” may be unclear. To clarify: our outputs are strictly structured as JSON blocks, as illustrated in Figure 1. Modern LLMs are highly capable of following these formatting instructions, with parsing failures occurring in <1% of cases. For example, across 30K+ questions, we observed the following JSON parsing error rates, which are all insignificant.
>   - GPT-4o: 72 (0.23%)
>   - o4-mini: 46 (0.15%)
>   - DeepSeek-R1: 47 (0.15%)
>   - LLaMA-3.3-70B: 168 (0.55%)
>
>
> We will clarify this point in the revision and include the exact statistics.
>
>
>
>
>
>
> ***Expert verification: What specific challenges of your standardization process warrant further human validation***
>
> - Expert verification plays a critical role throughout our benchmark construction and evaluation. Specifically, human reviewers are involved in: (1) Ensuring models interpret prompts as intended; (2) Verifying output adherence to the expected structured format; (3) Validating the correctness of our evaluation framework, etc. This human-in-the-loop process helps detect subtle issues and has been instrumental in refining both prompts and evaluation logic. We will include a detailed description of this process in the revised manuscript.
>
>
> ***Stochasticity of reasoning models, and the need to run benchmarks multiple times***
>
> - We appreciate the suggestion to examine model variance. We have conducted additional experiments, running reasoning models three times. For instance, o4-mini’s scores across runs were [0.639, 0.635, 0.637], confirming low stochastic variability. We will incorporate these findings into the revised version.
>
>
>
> ***The evaluation should also include Gemini models and TableGPT***
>
> - We thank the reviewer for this suggestion. In our updated results, we have expanded our evaluation to cover 14 additional models, including Gemini, TableGPT, LLaMA-4, Mistral, Qwen-3, Grok-3, and others. Below is a summary, with more details of these results included in our revised paper.
>
>
> | Model Type | Model                                       	| MMTU result 	|
> |------------|--------------------------------------------------|-----------------|
> | Chat   	| GPT-4o (2024-11-20)                          	| 0.491 ± 0.01	|
> |        	| Llama-4-Maverick-17B-128E-Instruct-FP8       	| 0.472 ± 0.01	|
> |        	| Llama-3.3-70B                                	| 0.438 ± 0.01	|
> |        	| Mistral-Large-2411                           	| 0.430 ± 0.01	|
> |        	| Mistral-Small-2503                           	| 0.402 ± 0.01	|
> |        	| GPT-4o-mini (2024-07-18)                     	| 0.386 ± 0.01	|
> |        	| Llama-4-Scout-17B-16E-Instruct               	| 0.377 ± 0.01	|
> |        	| Qwen3-32B (no thinking)                      	| 0.367 ± 0.01	|
> |        	| Qwen3-8B (no thinking)                       	| 0.343 ± 0.01	|
> |        	| TableGPT2-7B                                 	| 0.310 ± 0.01	|
> |        	| Qwen2.5-7B-Instruct                          	| 0.300 ± 0.01	|
> |        	| Llama-3.1-8B                                 	| 0.259 ± 0.01	|
> | Reasoning  | o4-mini (2024-07-18)                         	| **0.639 ± 0.01**|
> |        	| Gemini 2.5 Flash                             	| 0.606 ± 0.01	|
> |        	| Deepseek-R1                                  	| 0.596 ± 0.01	|
> |        	| Grok-3                                       	| 0.575 ± 0.01	|
> |        	| Qwen3-32B (thinking)                              	| 0.451 ± 0.01	|
> |        	| TAMA-Qwen3-8B                                	| 0.436 ± 0.01	|
> |        	| Qwen3-8B (thinking)                                  	| 0.428 ± 0.01	|
>
> These additions now provide a more comprehensive comparison across model types and capabilities.
>
>
> ***Comparing DeepSeek-R1 (~600B parameters) against LLaMA 70B is not a fair comparison.***
>
> - We acknowledge the concern regarding fairness in model scale comparisons. To address this, we included several direct comparisons between reasoning-enabled and standard chat models of the same base architecture. For example:
>   - Qwen3-32B (thinking): 0.451 vs. Qwen3-32B (no thinking): 0.368
>
>
>   - Qwen3-8B (thinking): 0.428 vs. Qwen3-8B (no thinking): 0.343
>
>
> These comparisons indeed help isolate the impact of reasoning augmentation independent of model size, which we will discuss in more detail in our revised paper.
>
>
> ***The error analysis section is appreciated, but currently too vague***
>
> - We appreciate the recognition of our initial error analysis and agree that providing additional details would be beneficial. We note that certain details (e.g., more concrete examples and illustrative screenshots) were omitted in the submitted paper in the interest of space, but are available in our technical report (linked in the submitted paper).
>
> - Like suggested, we will further expand our error analysis with fine-grained categorizations (e.g., breaking down table-understanding issues into sub-categories, such as column-alignment issues and long-context issues, together with illustrative examples), which we agree will make the error analysis more clear. We thank the reviewer for the helpful suggestion.

---

> > ### Comment · Reviewer_BG3j · 2025-08-04
> >
> > Thank you for your detailed response. I am raising my score from 2 to 3, as some clarifications have been provided. However, several key points remain unclear or insufficiently addressed:
> >
> > - W2 & W7: These concerns have not been addressed in the rebuttal. I kindly ask the authors to provide a direct response to these previously raised issues.
> >
> > - W3: This point was only partially addressed. I understand the authors prefer not to apply an LLM-based shallow filtering step. However, incorporating such a filtering step would not only improve data quality but also strengthen the benchmark’s novelty. It would distinguish the dataset by demonstrating not only its breadth but also a degree of post-processing rigor.
> >
> > - W4: While I understand the inclusion of certain information may be due to company policy, I question the value of including it in the paper. The space, along with page 8, could be better used for more constructive content. No clear justification has been provided for keeping this element in the manuscript.
> >
> > - W5: Due to the limited time available during the rebuttal phase, I understand a complete response may not be feasible. However, stating that "we will include a dedicated section in the revised manuscript" is not sufficient. Including even a draft or preliminary version of the proposed addition would have allowed me to evaluate its substance better.
> >
> > - W10: This point has not been adequately addressed. The rebuttal response repeats what is already written in the main paper (lines 173–177). The statement "we will include a detailed description…” should itself be substantiated in the rebuttal. More critically, my original concern remains: verifying only 20 samples per task results in approximately 0.017% of the dataset being human-verified, a proportion far too small to justify this step as a meaningful or reliable component of the pipeline. If the authors intend to retain this step, it should be significantly expanded or supplemented, e.g., through a lightweight LLM-based filtering procedure, ideally validated via a small-scale study demonstrating strong agreement between LLM outputs and expert annotations.
> >
> > - W14: The response does not specify what exactly will be added to the revised manuscript. A more concrete indication of the intended additions would help assess the authors’ commitment to addressing this feedback.

---

> > ### Author Response · Authors · 2025-08-08
> > **Response to Reviewer BG3j (regarding W2)**
> >
> > We thank the reviewer for their thoughtful engagement and detailed follow-up. During the discussion phase, we have made full use of the allotted time to address the points raised in the review:
> >
> >
> > ***W2. It remains unclear to what extent the review relies on keyword-based searches over paper titles, and what role the DeepResearch tool actually plays.***
> >
> > Thank you for the question. We provide more details on the 3 main steps we performed during task selection, and how the keyword/DeepResearch tools were used to assist the selection process as follows:
> >
> > - Our team of nine co-authors have years of experience working in complementary topics, such as table understanding, data integration, and data analysis, and are reasonably familiar with parts of the literature. We began by jointly brainstorming predictive table tasks known from the literature that satisfy the 3 requirements listed in Section 3.2 (real user facing, real-world datasets, and objectively evaluable). We arrived at an initial set of 21 tasks that satisfy all these requirements.
> >
> > - To ensure broader coverage and avoid blind spots, we systematically examined the DBLP pages of relevant conferences (e.g., SIGMOD, VLDB, PLDI, POPL, WWW, WSDM), from the last two decades year by year (e.g., https://dblp.org/db/conf/sigmod/sigmod2022.html for SIGMOD in year 2022). We performed keyword searches (e.g., “table,” “spreadsheet”) over these paper titles in each page, and manually reviewed the resulting matches to check against our three aforementioned requirements. This yielded 2 additional table tasks.
> >
> > - Lastly, since human keyword searches may miss relevant work, we used the DeepResearch tool in ChatGPT to scan the same set DBLP pages, providing a seed list of tasks meeting our requirements as examples, asking DeepResearch to go through each relevant paper to identify candidate tasks and datasets that meet our guidelines.  While DeepResearch returned many candidates, most failed to meet all of our requirements. After manual verification, we identified two further tasks this way.
> >
> > (A sketch of our prompt is shared below):
> >
> > """ Please check the last 20 years of {Conference_name} (e.g., SIGMOD/VLDB), starting from their DBLP pages, and give me a comprehensive list of papers that study predictive tasks on tables, that are:
> > -  have labeled benchmarks, based on real-world datasets
> > -  evaluated using predictive metrics, such as precision / recall
> >
> > Here is a list of papers from their DBLP pages:
> > {URLs_to_DBLP_pages}
> >
> > Here is a list of sample tasks that we are looking for, that meet our requirements:
> > {Sample_table_tasks_we_know}
> >
> > Please be comprehensive, and review each paper that may meet our requirements. Return your final results in a table that lists the name of the paper, the name of the task, and relevant benchmark datasets used in the evaluation. """
> >
> >
> > In this prompt, we replace {Conference_name} above with each conference of interest (e.g., SIGMOD/VLDB/PLDI, etc.), {URLs_to_DBLP_pages} lists relevant DBLP pages. DeepResearch usually comes back within an hour, listing the tasks/datasets it identifies as candidates, which we then perform further reviews. While this step returns many candidates, most would fail to meet all of our listed requirements.
> >
> > In the end, while keyword search and DeepResearch broadened our coverage, they contributed only marginally, adding 4 tasks beyond our initial expert-generated list. We will include these details in the appendix of our revised paper to facilitate replication and future work.

---

> > > ### Author Response · Authors · 2025-08-08
> > > **Response to Reviewer BG3j (regarding W3)**
> > >
> > > ***W3. Not all tasks may be completely "objectively evaluable," with ambiguity known in tasks like Text-to-SQL, due to the ambiguous nature of natural language. I recommend applying the same LLM-based shallow filtering to address this issue.***
> > >
> > > We appreciate the author for the recommendation. We agree that Text-to-SQL can indeed be ambiguous, like discussed in [1]. As suggested, we have performed a round of LLM-based shallow filtering using o4-mini in the past few days (which we found to perform better than GPT-4o on this filtering task). Here, we provide the model with each benchmark question, and ask LLM to determine whether there is ambiguity in the given question, and if so, return its answer in the following format:
> > >
> > > {“ambiguous“: true, “interpretation-1“: (interpretation), “interpretation-2“: (interpretation), “interpretation-3“: (interpretation), “explanation-of-ambiguity“: (your explanation)}.
> > >
> > > For example, for question bird_dev_20240627, LLM flags it as ambiguous in the following response, which we find to be reasonable:
> > >
> > > {"response":"{“ambiguous“: true, “interpretation-1“: “Find sets released between 2012-01-01 and 2015-12-31 in the 'sets' table, compute their average count per year, and then use 'set_translations' to identify which translation language occurs most often for those sets.“,  \n “interpretation-2“: “Find sets released between 2012-01-01 and 2015-12-31 in the 'sets' table, compute their average count per year, then join to 'cards' and 'foreign_data' to see which card translation language is most frequent for cards in those sets.“,  \n “explanation-of-ambiguity“: “The question refers to a “common language of the card" after asking about set releases; it could mean the most frequent translation language for the sets (from 'set_translations') or for the cards (from 'foreign_data').“}"
> > >
> > >
> > > After running the LLM-filtering step on each question across all tasks, we get the following result tabulated for each task, where tasks like NL2SQL indeed scores high on the list (the BIRD dataset has 18.5% cases labeled as ambiguous).
> > >
> > >
> > > | task                          	|   fraction_of_cases_labeled_ambiguious |
> > > |:----------------------------------|------------------:|
> > > | NL2SQL                        	|    	0.131651   |
> > > | Table-QA                      	|    	0.125  	|
> > > | header-value-matching         	|    	0.0546218  |
> > > | Columns-property-anotation    	|    	0.047  	|
> > > | Functional-Dependency         	|    	0.0453074  |
> > > | List-to-table                 	|    	0.043  	|
> > > | Table-Fact-Verification       	|    	0.041  	|
> > > | Data-transform-reshape        	|    	0.0336134  |
> > > | Error-Detect                  	|    	0.0327126  |
> > > | Formula-prediction-context    	|    	0.0311638  |
> > > | Column-type-annotation        	|    	0.027  	|
> > > | Transform-by-output-target-schema |    	0.0214286  |
> > > | equi-join-detect              	|    	0.0193424  |
> > > | Data-transform-pbe            	|    	0.0176056  |
> > > | Data-Imputation               	|    	0.0145 	|
> > > | Transform-by-input-output-table   |    	0.0114943  |
> > > | Cell-entity-annotation        	|    	0.009  	|
> > > | semantic-join                 	|    	0.00763359 |
> > > | Entity-Matching               	|    	0.00616438 |
> > > | Schema-Matching               	|    	0.0055325  |
> > > | String-Relationship           	|    	0.00130548 |
> > > | Arithmetic-Relationship       	|    	0.001221   |
> > > | Table-needle-in-a-haystack    	|    	0.001  	|
> > > | Table-Locate-by-Row-Col       	|    	0      	|
> > > | semantic-transform            	|    	0      	|
> > >
> > >
> > >
> > > After removing all these flagged questions (1255, or 4% of total), the resulting MMTU scores of 4 main models compared in the paper show a very small change (less than 1%), like shown below.
> > >
> > >
> > > | Model        |   MMTU score (original)  |  MMTU score (after ambiguity filtering)   |
> > > |:-------------|---------:|---------:|
> > > | o4-mini      | 0.63941  | 0.64314  |
> > > | Deepseek-R1  | 0.59621  | 0.600669 |
> > > | GPT-4o       | 0.49052  | 0.494792 |
> > > | Llama33-70B  | 0.437973 | 0.442314 |
> > >
> > >
> > > We will incorporate this filtering step into our work, and update our results in the paper accordingly.
> > >
> > >
> > >
> > >
> > > [1] Saparina, Irina, and Mirella Lapata. "Ambrosia: A benchmark for parsing ambiguous questions into database queries." Advances in Neural Information Processing Systems 37 (2024): 90600-90628.

---

> > ### Author Response · Authors · 2025-08-08
> > **Response to Reviewer BG3j (regarding W4)**
> >
> > ***W4. The rationale for filtering datasets based on privacy or security is not clearly stated in the paper.***
> >
> > We agree with the reviewer that the privacy filtering step could benefit from additional detail. Although all our datasets originate strictly from the public domain, we treated them as if we were “republishing” and therefore took extra precautions. Specifically, we applied the following prompt for filtering. This process flagged a small fraction (~2%) of the data (some of which were false positives), but we nonetheless removed all flagged portions out of an abundance of caution.
> >
> >
> > The prompt used for filtering:
> >
> > """I am preparing a dataset for public release. The dataset consists of publicly available data (e.g., public benchmarks, web tables, etc.). Please review the dataset, and ensure that it does not contain sensitive private information, such as ssn, credit card information, Bank account number, Driver's license number, etc. (Note that because the data is from the public domain, names and addresses are considered acceptable.)
> >
> > If you believe the text contains sensitive or PII information, return {"PII": "yes"}, otherwise return {"PII": "no"}. No explanation is needed.
> >
> > {{{question}}}"""
> >
> > We plan to merge the description of this shallow filtering step, together with the ambiguity-based shallow LLM filtering (W3) and quality-based shallow filtering (W5), together into a new step. We will describe this expanded step in Section 3.2 of our revised manuscript, based on reviewer’s suggestions.

---

> > ### Author Response · Authors · 2025-08-08
> > **Response to Reviewer BG3j (regarding W5 - part 1)**
> >
> > ***W5. Detailed statistics on each dataset's contributions or the level of data curation applied. Have the datasets been manually checked for errors or inconsistencies? For example, BIRD is known to have limitations, which should be acknowledged.***
> >
> >
> > We provide detailed statistics of each dataset’s contribution to our overall benchmark as follows (which we will add to the appendix of our revised paper).
> >
> > | task                          	| dataset          	|   count |
> > |:----------------------------------|:---------------------|--------:|
> > | Arithmetic-Relationship       	| Auto-Relate      	| 	819 |
> > | Cell-entity-annotation        	| SemTab2019       	|	1000 |
> > | Column-type-annotation        	| SemTab2019       	|	1000 |
> > | Columns-property-anotation    	| SemTab2019       	|	1000 |
> > | Data-Imputation               	| WebTable         	|	1000 |
> > | Data-Imputation               	| tablib           	|	1000 |
> > | Data-transform-pbe            	| TDE              	| 	236 |
> > | Data-transform-pbe            	| Transformation-text  | 	332 |
> > | Data-transform-reshape        	| Auto-Tables      	| 	238 |
> > | Entity-Matching               	| Amazon-Google    	|	1000 |
> > | Entity-Matching               	| BeerAdvo-RateBeer	|  	88 |
> > | Entity-Matching               	| DBLP-ACM         	|	1000 |
> > | Entity-Matching               	| DBLP-Scholar     	|	1000 |
> > | Entity-Matching               	| Fodors-Zagats    	| 	186 |
> > | Entity-Matching               	| Walmart-Amazon   	|	1000 |
> > | Entity-Matching               	| iTunes-Amazon    	| 	106 |
> > | Error-Detect                  	| Relational-Tables	| 	999 |
> > | Error-Detect                  	| Spreadsheet-Tables   | 	988 |
> > | Formula-prediction-context    	| cisco-random     	| 	897 |
> > | Formula-prediction-context    	| enron-random     	| 	763 |
> > | Formula-prediction-context    	| pge-random       	| 	967 |
> > | Formula-prediction-context    	| ti-random        	| 	999 |
> > | Functional-Dependency         	| Auto-Relate      	| 	309 |
> > | List-to-table                 	| TEGRA            	|	1000 |
> > | NL2SQL                        	| Archer           	| 	104 |
> > | NL2SQL                        	| KaggleDBQA       	| 	185 |
> > | NL2SQL                        	| Spider           	|	1000 |
> > | NL2SQL                        	| WikiSQL          	|	1000 |
> > | NL2SQL                        	| bird             	|	1000 |
> > | Schema-Matching               	| DeepMDatasets    	|   	7 |
> > | Schema-Matching               	| HXD              	| 	172 |
> > | Schema-Matching               	| Wikidata         	|   	4 |
> > | Schema-Matching               	| assays           	| 	180 |
> > | Schema-Matching               	| miller2          	| 	180 |
> > | Schema-Matching               	| prospect         	| 	180 |
> > | String-Relationship           	| Auto-Relate      	| 	766 |
> > | Table-Fact-Verification       	| TabFact          	|	1000 |
> > | Table-Locate-by-Row-Col       	| Tablib      	|	1000 |
> > | Table-QA                      	| FinQA            	|	1000 |
> > | Table-QA                      	| TableBench       	| 	424 |
> > | Table-QA                      	| WikiQA           	|	1000 |
> > | Table-needle-in-a-haystack    	| Tablib     	|	1000 |
> > | Transform-by-input-output-table   | AutoPandas       	|  	27 |
> > | Transform-by-input-output-table   | Scythe           	|  	60 |
> > | Transform-by-output-target-schema | commercial-pipelines |  	13 |
> > | Transform-by-output-target-schema | github-pipelines 	| 	687 |
> > | equi-join-detect              	| Auto-BI          	| 	517 |
> > | header-value-matching         	| TableGPT         	| 	952 |
> > | semantic-join                 	| DataXFormer      	|  	81 |
> > | semantic-join                 	| SEMA-join        	|  	50 |
> > | semantic-transform            	| DataXFormer      	|  	81 |
> > | semantic-transform            	| SEMA-join        	|  	50 |
> >
> > (due to space constraint, to be continued in the next comment)

---

> > > ### Author Response · Authors · 2025-08-08
> > > **Response to Reviewer BG3j (regarding W5 - part 2)**
> > >
> > > While we have sampled 20 examples per task for manual checking, during the rebuttal phase, we have performed an additional LLM-based quality check, by prompting LLM (o4-mini) to check each question and its ground-truth for quality, and then mark the ground-truth answer as either correct or incorrect (note that some tasks have objective ground-truth by construction, e.g. for data imputation, the ground-truth is naturally the cell that we remove from the original table, and we do not subject such tasks to LLM-based quality checks).
> > >
> > > The list of tasks with questions that are flagged with quality issues are listed below:
> > >
> > >
> > > | task                       |   fraction of cases with quality_label=True  |  fraction of cases with  quality_label=False |
> > > |:---------------------------|----------------------:|----------------------:|
> > > | Table-QA                   |     0.828383          |     0.171617          |
> > > | NL2SQL                     |     0.844026          |     0.155974          |
> > > | Cell-entity-annotation     |     0.899              |     0.101              |
> > > | Functional-Dependency      |     0.899676           |     0.100324           |
> > > | Column-type-annotation     |     0.902              |     0.098              |
> > > | Columns-property-anotation |     0.915              |     0.085              |
> > > | Schema-Matching            |     0.929461           |     0.070539           |
> > > | Table-Fact-Verification    |     0.946              |     0.054              |
> > > | Entity-Matching            |     0.970776           |     0.029224           |
> > > | equi-join-detect           |     0.972921           |     0.027079           |
> > > | Error-Detect               |     0.983895           |     0.016105           |
> > >
> > >
> > >
> > >
> > > We note that despite our repeated attempt to optimize our prompt, we observe LLM can still be overly confident and can over-trigger on some (question, ground-truth) pairs, by flagging them as “incorrect ground-truth” (as the model itself may have blind spots in its knowledge). We therefore performed another round of manual checking, by manually checking 10 samples per task with  quality_label=True  and quality_label=False.
> > >
> > > We found that in cases where quality_label=True, human experts agree 100% with the LLM – therefore confirming that LLM and human experts do agree, that the vast majority of our benchmark (over 95% of the questions) are indeed correct.
> > >
> > > On the remaining cases where  quality_label=False, (shown in the second column of the table above), secondary human experts verification shows that humans agree 50%-60% of the time with LLM, as when LLM’s label of some ground-truth to be incorrect, may actually be blind spots in its knowledge (e.g., in CTA tasks, LLM will judge a ground-truth DBPedia link as incorrect when it is in fact correct).
> > >
> > > Nevertheless, to ensure result quality, we take the conservative approach of removing all questions that are flagged as “ground-truth incorrect” by LLM (5% of all questions). After removing these questions, our overall results show a small, 1-2% improvement for all models, like listed below. Removing all questions flagged as ambiguous (discussed in W3) gives a further ~0.5% boost for each model.
> > >
> > >
> > > | Model        |   MMTU (original)   |  MMTU (after quality filtering) |  MMTU (after quality and ambiguity filtering)   |
> > > |:-------------|---------:|---------:|-----------:|
> > > | o4-mini      | 0.63941  | 0.65707  | 0.659871   |
> > > | Deepseek-R1  | 0.59621  | 0.612994 | 0.617125   |
> > > | GPT-4o       | 0.49052  | 0.503231 | 0.507329   |
> > > | Llama33-70B  | 0.437973 | 0.450267 | 0.454347   |
> > >
> > > While the overall effect of the filtering steps are small, we plan to update results in our paper with these filterings, which we agree can further improve the quality of our benchmark.

---

> > > > ### Author Response · Authors · 2025-08-08
> > > > **Response to Reviewer BG3j (regarding W7)**
> > > >
> > > > ***W7. Code implementation for Execution Accuracy seems sensitive to column orders.***
> > > >
> > > >
> > > > We thank the reviewer for the very careful analysis of our code, and we agree that column order does not to need to be strictly enforced for Execution Accuracy in Text-2-SQL, and we have added two lines to the Text-2-SQL evaluator function at Line 20 [4] like below, to ignore the column order:
> > > >
> > > >  ```python
> > > > list1[i] = sorted(list1[i], key=lambda x: str(x).lower())
> > > > list2[i] = sorted(list2[i], key=lambda x: str(x).lower())
> > > >  ```
> > > >
> > > > This two line change leads to a small improvement for all models (less than 1% in Text-2-SQL, and less than 0.04% overall), like shown below. We will update our paper accordingly using these new results.
> > > >
> > > >
> > > > | Model | accuracy on Text-2-SQL (before)    | accuracy  on Text-2-SQL (after) |
> > > > |----------|---------|---------|
> > > > | O4-mini | 0.587683 | 0.596282 |
> > > > | Deep seek-R1 | 0.561541 | 0.570141 |
> > > > | GPT-4o | 0.477286 | 0.486286 |
> > > > | Llama33-70b | 0.493748** | 0.503748 |
> > > >
> > > >
> > > >
> > > > We would like to mention that this two line change serves as an example to show the extensibility of our benchmarking framework (discussed in our earlier response to the reviewer’s W6) – with a few lines we can similarly implement additional evaluation metrics such as Qatch [3], for Text-2-SQL or other additional tasks.
> > > >
> > > > [3] Papicchio, Simone, Paolo Papotti, and Luca Cagliero. "Qatch: Benchmarking sql-centric tasks with table representation learning models on your data." Advances in Neural Information Processing Systems 36 (2023): 30898-30917.
> > > >
> > > > [4] https://github.com/MMTU-Benchmark/MMTU/blob/04890ae184fe9773a8b7297902c7ea4818c0a5ac/evaluators/nl2sql.py#L23

---

> > ### Author Response · Authors · 2025-08-08
> > **Response to Reviewer BG3j (regarding W10)**
> >
> > ***W10. The "Expert verification" step may be insufficient to draw reliable conclusions. It is recommended to either remove the step, or significantly expand the verification.***
> >
> >
> > - Thank you for the feedback. Following your suggestion in W5, we have expanded the verification step using LLM-based filtering, by prompting LLM to inspect each question and its ground-truth answer, to flag questions for which LLM believes the ground-truth to be incorrect.
> >
> > - Out of the 30647 questions, LLM marked 29124 as “ground-truth correct”, and 1523 as “ground-truth incorrect”. We then used human experts to perform manual verification, on 20 questions per task for questions that are marked as  “ground-truth correct” and  “ground-truth incorrect”.
> >
> > - On the sampled verification, we found 100% human-LLM agreement on the subset flagged as “ground-truth correct”, and 50%-60% human-LLM agreement on the subset flagged as “ground-truth incorrect” (as the LLM sometimes have knowledge gaps and can make mistakes itself). We therefore retain the 29124 questions labeled as “ground-truth correct” (95% of the original set) as our final question set (since  human-LLM have 100% agreement on their correctness on sampled evaluation). Please find details of this step in our response to (W5 - part 2) above, and we will update our manuscript to reflect these changes.

---

> > > ### Author Response · Authors · 2025-08-08
> > > **Response to Reviewer BG3j (regarding W14 - part 1)**
> > >
> > > ***W14. The error analysis section is appreciated, but currently too vague.***
> > >
> > > Our error analysis reported in the submitted paper is based on errors produced in reasoning models (o4-mini). We appreciate the suggestion and have updated our error analysis as follows. We first built a detailed categorization of typical errors we observe in models’ answers, using the following 5 super-categories, and 14 sub-categories:
> > >
> > > **Super-category: Table‐Understanding Errors**
> > > - Row/Column index misalignment (e.g., the row/column index is not correctly recognized in a table, leading to incorrect model output)
> > > - Long context (e.g., long table causes model to produce incorrect results)
> > > - Multi-table understanding (e.g., join relationships between multiple tables are identified incorrectly, leading to incorrect answers)
> > > - Complex table (e.g., merged cells spanning multi-row/cols, hierarchical headers, etc., causing models to produce incorrect answers)
> > >
> > > **Super-category: Reasoning & Coding Errors**
> > > - Syntax or execution error (if the task ask for a SQL/Python code, and the code should encounter execution error)
> > > - Semantically incorrect code (if the task ask for a SQL/Python code, and the code is executable but the result is incorrect)
> > > - Incorrect reasoning (pick this category only if the model's reasoning shows obvious flaws that directly lead to incorrect answers)
> > >
> > >
> > >
> > > **Super-category: Knowledge Errors**
> > > - Hallucinated facts (e.g., in CEA, output knowledge base references that do not exist)
> > > - Incorrect knowledge (e.g., in data imputation, filled in a related but incorrect value)
> > > - Numerical Errors (e.g., in data imputation, filled a numerical value that is close but not precise)
> > >
> > > **Super-category: Other Errors**
> > > - Format error / Result‐extraction failure
> > > - Timeout
> > > - Ground‐truth ambiguity / quality
> > > - None of the above: create a new category, and give a short description
> > >
> > >
> > >
> > > The distribution of errors (in both super-categories and sub-categories) on all 11921 questions for which o4-mini was incorrect can be found below. We observe that reasoning & coding, and table understanding remain two largest categories of errors.
> > >
> > > | o4-mini: super_categories             	|   	count |
> > >  |:---------------------------------|------------:|
> > >  | Reasoning & Coding Errors    	| 0.40347 	|
> > >  | Table-Understanding Errors   	| 0.256496	|
> > >  | Knowledge Errors             	| 0.163711	|
> > >  | Other Errors                 	| 0.117472	|
> > >
> > >
> > >
> > > | o4-mini: super_categories       	| categories                           	|  	count |
> > > |:---------------------------|:-----------------------------------------|-----------:|
> > > | Reasoning & Coding Errors  | Incorrect reasoning                  	| 0.226307   |
> > > | Table-Understanding Errors | Row/Column index misalignment        	| 0.177851   |
> > > | Reasoning & Coding Errors  | Semantically incorrect code          	| 0.173876   |
> > > | Knowledge Errors       	| Incorrect knowledge                  	| 0.134439   |
> > > | Other Errors           	| Ground-truth ambiguity / quality     	| 0.0582391  |
> > > | Other Errors           	| Format error / Result-extraction failure | 0.0405075  |
> > > | Knowledge Errors       	| Hallucinated facts                   	| 0.0230816  |
> > > | Table-Understanding Errors | Multi-table understanding            	| 0.0182666  |
> > > | Table-Understanding Errors | Complex table                        	| 0.0171201  |
> > > | Table-Understanding Errors | Long context                         	| 0.00848364 |
> > > | Knowledge Errors       	| Numerical Errors                     	| 0.00542648 |
> > > | Table-Understanding Errors | Column semantic misinterpretation    	| 0.00168144 |
> > > | Reasoning & Coding Errors  | Syntax or execution error            	| 0.00152858 |
> > > | Table-Understanding Errors | Incorrect join inference             	| 0.00122287 |
> > > | Other Errors           	| Incorrect evaluation                 	| 0.00114644 |
> > > | Table-Understanding Errors | Incorrect column matching            	| 0.00114644 |
> > > | Other Errors           	| Spurious relationship detection      	| 0.00099358 |
> > > | Other Errors           	| False positive error detection       	| 0.00099358 |
> > > | Other Errors           	| Evaluation error                     	| 0.00099358 |

---

> > > > ### Author Response · Authors · 2025-08-08
> > > > **Response to Reviewer BG3j (regarding W14 - part 2)**
> > > >
> > > > Similar results are observed for R1, GPT-4o, and other models, like shown below.
> > > >
> > > >
> > > > | Deepseek-R1: super_categories             	|   	count |
> > > >  |:---------------------------------|------------:|
> > > >  | Reasoning & Coding Errors    	| 0.452904	|
> > > >  | Table-Understanding Errors   	| 0.221796	|
> > > >  | Other Errors                 	| 0.19073 	|
> > > >  | Knowledge Errors             	| 0.103647	|
> > > >
> > > >
> > > > | Deepseek-R1: super_categories       	| categories                               	|   	count |
> > > > |:---------------------------|:---------------------------------------------|------------:|
> > > > | Reasoning & Coding Errors  | Incorrect reasoning                      	| 0.37428 	|
> > > > | Table-Understanding Errors | Row/Column index misalignment            	| 0.15547 	|
> > > > | Other Errors           	| Format error / Result-extraction failure 	| 0.101514	|
> > > > | Knowledge Errors       	| Incorrect knowledge                      	| 0.0835288   |
> > > > | Reasoning & Coding Errors  | Semantically incorrect code              	| 0.0762778   |
> > > > | Other Errors           	| Timeout                                  	| 0.0577948   |
> > > > | Other Errors           	| Ground-truth ambiguity / quality         	| 0.028933	|
> > > > | Table-Understanding Errors | Complex table                            	| 0.0163503   |
> > > > | Table-Understanding Errors | Multi-table understanding                	| 0.0162792   |
> > > > | Knowledge Errors       	| Hallucinated facts                       	| 0.0155684   |
> > > > | Knowledge Errors       	| Numerical Errors                         	| 0.00433639  |
> > > > | Reasoning & Coding Errors  | Syntax or execution error                	| 0.00220374  |
> > > > | Table-Understanding Errors | Long context                             	| 0.0018483   |
> > > > | Table-Understanding Errors | Column semantic misinterpretation        	| 0.00170612  |
> > > > | Table-Understanding Errors | Column semantics misinterpretation       	| 0.00163503  |
> > > > | Table-Understanding Errors | Column misinterpretation                 	| 0.000781972 |
> > > > | Table-Understanding Errors | Column semantics misunderstanding        	| 0.00042653  |
> > > > | Table-Understanding Errors | Incorrect join inference                 	| 0.00042653  |
> > > > | Table-Understanding Errors | Column header abbreviation misinterpretation | 0.00042653  |
> > > > | Table-Understanding Errors | Misinterpretation of column semantics    	| 0.000355442 |
> > > >
> > > > | GPT-4o: super_categories                     	|   	count |
> > > >  |:-----------------------------------------|------------:|
> > > >  | Reasoning & Coding Errors            	| 0.367047	|
> > > >  | Table-Understanding Errors           	| 0.35668 	|
> > > >  | Knowledge Errors                     	| 0.153533	|
> > > >  | Other Errors                         	| 0.0768898   |
> > > >
> > > >
> > > >
> > > > |  GPT-4o: super_categories       	| categories                           	|   	count |
> > > >  |:---------------------------|:-----------------------------------------|------------:|
> > > >  | Table-Understanding Errors | Row/Column index misalignment        	| 0.283122	|
> > > >  | Reasoning & Coding Errors  | Incorrect reasoning                  	| 0.223264	|
> > > >  | Reasoning & Coding Errors  | Semantically incorrect code          	| 0.138167	|
> > > >  | Knowledge Errors       	| Incorrect knowledge                  	| 0.105523	|
> > > >  | Knowledge Errors       	| Hallucinated facts                   	| 0.0334465   |
> > > >  | Other Errors           	| Format error / Result-extraction failure | 0.0323357   |
> > > >  | Other Errors           	| Ground-truth ambiguity / quality     	| 0.0323357   |
> > > >  | Table-Understanding Errors | Complex table                        	| 0.0156742   |
> > > >  | Table-Understanding Errors | Multi-table understanding            	| 0.0153039   |
> > > >  | Knowledge Errors       	| Numerical Errors                     	| 0.014008	|
> > > >  | Table-Understanding Errors | Long context                         	| 0.00715828  |
> > > >  | Reasoning & Coding Errors  | Syntax or execution error            	| 0.00456649  |
> > > >  | Table-Understanding Errors | Column semantic misinterpretation    	| 0.00203641  |
> > > >  | Table-Understanding Errors | Column semantics misinterpretation   	| 0.00098735  |
> > > >  | Table-Understanding Errors | Incorrect column matching            	| 0.000863931 |
> > > >  | Other Errors           	| Evaluation error                     	| 0.000863931 |
> > > >  | Table-Understanding Errors | Incorrect column mapping             	| 0.000678803 |
> > > >  | Table-Understanding Errors | Incorrect join inference             	| 0.000678803 |
> > > >  | Table-Understanding Errors | Incorrect join identification        	| 0.000678803 |
> > > >  | Other Errors           	| Incorrect evaluation                 	| 0.000617093 |
> > > >
> > > >
> > > > We will present these additional details, as well as exemplar cases from each error category, in our updated manuscript, which we agree will strengthen the error analysis of our paper.
> > > >
> > > > Please let us know if there are any further details we can clarify. We appreciate your thoughtful feedback and active engagement in our discussions!

---

> > > > > ### Comment · Reviewer_BG3j · 2025-08-08
> > > > >
> > > > > I thank the authors for the really detailed answer. I am changing my score accordingly. Well done!

---

> > > > > > ### Author Response · Authors · 2025-08-08
> > > > > >
> > > > > > Dear Reviewer BG3j,
> > > > > >
> > > > > > We sincerely appreciate your constructive feedback and valuable guidance in helping us improve our work, thank you!

---

### Official Review · Reviewer_wEH9 · 2025-06-25

**Rating:** 4
**Confidence:** 3

**Summary:**

The paper introduces MMTU, a large-scale benchmark specifically designed to evaluate the capabilities of large language models (LLMs) on complex, real-world table-centric tasks. Unlike prior table benchmarks, MMTU comprehensively covers 25 diverse table-related tasks, reflecting challenges faced by professional users such as data analysts, engineers, and database administrators.

- Scale and Diversity: MMTU comprises 30,647 questions over 67,886 real tables, sourced from 52 datasets.

- Broad Task Coverage: The benchmark includes tasks such as table transformation, schema and entity matching, data cleaning, table joins, column transformations, arithmetic and functional relationships, table understanding, NL-to-SQL, table QA, and knowledge base mapping.

- Real-World Relevance:: All are expert-level, real-world tasks with objective ground truths, derived from web tables, spreadsheets, and relational databases.

**Dataset Code Accessibility:**

Yes

**Dataset Code Comments:**

The dataset is available on Hugging Face, as uploaded by the authors

**Ethical Comments:**

There are no ethical concerns.

**Ethical Considerations:**

No, there are no or only very minor ethics concerns

**Final Justification:**

I've reviewed detailed rebuttal and the points that author made in the paper's appendices. Authors explanations have addressed my primary concern regarding the benchmark's scientific novelty. The findings on LLMs' performance with long-context tables and semantically-invariant perturbations suggest valuable, non-obvious insights into their structural understanding. While I still have some reservations, authors clarification of these specific insights demonstrates that the work contributes more than a simple aggregation of datasets. Accordingly, I have made an adjustment to my score.

**Limitations Weaknesses:**

1. The benchmark is constructed by aggregating a wide range of existing datasets and tasks, but it is not clear what the key novelty of this integration is. Simply combining prior datasets, while practically useful, may not be sufficient as a research contribution
2. Relatedly, the analysis section does not seem to reveal any particularly novel findings that emerge from unifying these datasets. The paper largely reports performance scores in a descriptive manner, but lacks a strong central claim or insight that would motivate the importance of this benchmark from a scientific perspective.

**Strengths Contributions:**

1. Comprehensive Coverage of Real-World Table Tasks:
The MMTU benchmark goes far beyond existing table evaluation benchmarks by covering 25 diverse, real-world tasks that professional users such as data analysts and engineers actually face.

2. Robust and Carefully Curated Dataset:
The benchmark is built on 30,647 questions derived from 52 datasets and 67,886 real tables, ensuring broad coverage and high data diversity.

3. Detailed Evaluation Framework and Analysis
The paper provides a sophisticated and extensible evaluation framework that supports complex output formats like SQL, Python, and structured JSON, allowing for execution-based and structured evaluations rather than simple string matching.

---

> ### Author Rebuttal · Authors · 2025-07-31
>
> We thank the reviewer for recognizing the scale, diversity, and comprehensiveness of our proposed benchmark. We address the weaknesses raised in the review as follows:
>
> ***The benchmark is constructed by aggregating a wide range of existing datasets and tasks, but it is not clear what the key novelty of this integration is***.
>
> Thank you for raising this important point. We would like to clarify the key novelties of our benchmark:
>
> - **Substantially broader Coverage of Table Tasks**: While existing LLM evaluations predominantly focus on Text-to-SQL and Table QA, our work is the first to recognize and address the broader landscape of table-related tasks that are critical in real-world settings but largely studied outside the ML/NLP community (e.g., SIGMOD/VLDB, PLDI/POPL). In contrast to NLP and multi-modal model evaluation, where there is an abundance of comprehensive benchmarks (MMLU, MMMU, Big-Bench, etc.), there has been no similarly comprehensive benchmark for table tasks, making this novel as a new “dataset and benchmark” contribution.
>
> - **Novel Integration for LLM Evaluation**: We curate and standardize over 30K examples spanning 25 distinct table tasks and 52 benchmark datasets, and design a corresponding benchmarking framework that facilitates the execution, evaluation, and analysis of these diverse tasks within a single framework. We should note that, out of the 25 tasks and 52 benchmark datasets, only 6 tasks have been previously used in LLM evaluation (which are mainly Text-2-SQL and Table-QA tasks), making our dataset novel for LLM evaluation and new for the ML/NLP community.
>
>
> - **A New Testbed for Reasoning Models**: While reasoning models are quickly gaining popularity, they are typically benchmarked on math and code tasks. We show for the first time using our comprehensive table benchmarks, that table tasks can also benefit significantly from reasoning. This positions our benchmark as a new, complementary testbed for evaluating and advancing reasoning capabilities in LLMs, on top of math / coding tasks that are currently used to evaluate reasoning models.
>
> - **New Empirical Insights**: Beyond the benchmark construction, we produce new findings from detailed analysis of running the new benchmark, which we detail in our response to the next reviewer comment.
>
>
> ***Relatedly, the analysis section does not seem to reveal any particularly novel findings that emerge from unifying these datasets.***
>
> - We appreciate the reviewer’s feedback and would like to respectfully clarify the key insights that emerge from our unified and comprehensive benchmark evaluation (We invite the reviewer to check Section 4 and Appendices C/D of the paper for full details). Notable findings include:
>
>   - **LLMs show strong potential on tabular tasks**, with newer and larger models consistently outperforming older and smaller ones. This trend underscores the promise of leveraging frontier LLMs for table and data tasks. (Section 4).
>
>   - **Reasoning-oriented models significantly outperform general-purpose chat models** (e.g., o4-mini vs. GPT-4o, or Qwen-3-32B-thinking vs. Qwen-3-32B-no-thinking), with over 10 percentage points improvement. This highlights the potential of leveraging reasoning models for table tasks, as well as making our table benchmark a new and valuable testbed for evaluating advancing reasoning capabilities in LLMs (Table 3, Section 4.1).
>
>
>   - **Modern LLMs exhibit robustness to input table format**, in contrast to earlier models that were sensitive to serialization styles (e.g., Markdown vs. CSV vs. JSON vs. HTML). This finding suggests reduced need for prompt engineering and reflects broader progress in table comprehension (Figure 8, Section 4.2).
>
>   - **Long-context table inputs remain a key challenge**: LLMs struggle with large tables containing many rows and columns, echoing past limitations observed in long-context NLP tasks. This points to an important direction for future research (Figure 6, Section 4.2; Figure 11, Appendix D).
>
>   - **LLM performance degrades under semantically-invariant table perturbations**, such as row/column shuffling. This indicates current models may lack true structural understanding of tables and points to possible opportunities for architectural or training improvements (Figure 7, Section 4.2).
>
>
> We hope these insights help illustrate the value of our unified benchmark and the novel empirical findings it enables, and we invite the reviewer to check our paper, particularly Section 4 and Appendices C/D for full details of our findings.

---

> > ### Comment · Reviewer_wEH9 · 2025-08-04
> > **Official Comment**
> >
> > Thank you for the detailed response. While I acknowledge that the proposed benchmark is a very valuable 'resource' for the community, my core concern regarding its 'scientific novelty' has not been resolved. The presented analytical findings seem to reconfirm known characteristics of LLMs in a new domain, rather than providing novel scientific insights.

---

> > > ### Comment · Area_Chair_TNaB · 2025-08-07
> > >
> > > Dear Reviewer wEH9,
> > >
> > > Thanks for reviewing the paper and for responding to author rebuttal.
> > >
> > > For concrete discussion, can you provide more details and justification, with specific references and contrast to prior work and the authors' points, for your concern on novelty?
> > >
> > > Thanks!

---

> > ### Author Response · Authors · 2025-08-07
> >
> > Dear reviewer wEH9,
> >
> > First of all, we thank you again for your kind recognition of our work’s contributions — including its “Scale and Diversity,” “Broad Task Coverage,” “Real-World Relevance,” “Robust and Carefully Curated Dataset,” and “Detailed Evaluation Framework and Analysis.” We are truly grateful for your encouraging feedback.
> >
> > Regarding your central concern about the “novel scientific findings” of our work, and whether our presented results only “reconfirm known characteristics of LLMs in a new domain” – we respectfully wish to clarify that our study surfaces several key findings that, to the best of our knowledge, are either _novel or challenging prevailing assumptions_. These findings are made possible precisely because of the scale and diversity of our benchmark:
> >
> > - ***Long-context table inputs remain unsolved for state-of-the-art LLMs***: While frontier models like GPT-4o and Llama 4 perform near-perfectly on long-context _textual_ tasks (e.g., needle-in-a-haystack [3, 103, 90, 26, 14]; see our Figure 10 with perfect “green” results), we show that their performance plummets, sometimes below 20%, on the analogous _table-based_ long-context tasks we introduce (see Figure 11, which is almost all “red”). This stark contrast, not previously reported, challenges  the common belief that “long-context reasoning is a solved problem” [103], suggesting that progress in textual document modeling may not directly transfer to structured data like tables. Our results thus highlight a critical and underexplored gap in current LLM capabilities for future research.
> >
> >
> > - ***Modern LLMs exhibit robustness to input table format***: Prior work has shown that earlier LLMs (e.g., GPT-3.5 and GPT-4) were highly sensitive to table serialization formats (Markdown vs. CSV vs. JSON vs. HTML) [88, 110]. In contrast, we find that newer models exhibit strong robustness across formats (Figure 8), which is a positive new development not widely known or documented using systematic evaluations. This can reduce the need for prompt engineering and table format optimization for future research in the area of LLM for tables.
> >
> >
> > - ***LLM performance degrades under semantically-invariant table perturbations***: Despite their robustness to table formats (markdown vs. JSON), current models still suffer significant performance drops under semantically-preserving table transformations like row or column shuffling, which should not alter a table’s meaning for humans (Figure 7). This suggests that true structural understanding of tabular data remains an open challenge, pointing to opportunities for improvements in model architectural or training.
> >
> >
> > We acknowledge that detailed findings like these may have been somewhat buried in the current paper (e.g., some are presented in later sections or even in the appendix), due to space limitations, as our paper's primary goal is to describe our new large-scale benchmark and how it is constructed, which inadvertently pushed some of our key findings into later sections of the paper. We thank the reviewer for the suggestion, and will make sure to better highlight these key insights in our revised manuscript.
> >
> > Finally, much like earlier benchmark efforts such as MMLU and MMMU, we hope our benchmark will serve as a powerful catalyst for future scientific advancements. While this paper presents several important findings, our primary goal for submitting to the Benchmarks and Datasets track is to empower the broader research community to uncover many more novel scientific insights in this important yet under-explored area, which has thus far lacked a systematic benchmark, an objective we believe is fully aligned with the goals of the Benchmarks and Datasets track.
> >
> >
> >
> >
> > **References**: (using the same numbering as in the submitted paper)
> >
> > [3] LLM test: Needle in a haystack. https://github.com/gkamradt/LLMTest_NeedleInAHaystack.
> >
> > [14] Claude 3 release announcement. https://www.anthropic.com/news/claude-3-family
> >
> > [26] llama 4 announcement. https:
> > //ai.meta.com/blog/llama-4-multimodal-intelligence/
> >
> > [88] Peng Li et al. Table-gpt: Table-tuned gpt for diverse table
> > tasks. arXiv preprint arXiv:2310.09263, 2023.
> >
> > [90] Aixin Liu, et al. Deepseek-v3 technical report. arXiv
> > preprint arXiv:2412.19437, 2024.
> >
> > [103] Jonathan Roberts, Kai Han, and Samuel Albanie. Needle threading: Can llms follow thread through near-million-scale haystacks? arXiv preprint arXiv:2411.05000, 2024
> >
> > [110] Yuan Sui, Mengyu Zhou, Mingjie Zhou, Shi Han, and Dongmei Zhang. Table meets llm: Can large language models understand structured table data? a benchmark and empirical study. In WSDM, 2024.
> >
> > [111] Gemini Team, Petko Georgiev, Ving Ian Lei, Ryan Burnell, Libin Bai, Anmol Gulati, Garrett
> > Tanzer, Damien Vincent, Zhufeng Pan, Shibo Wang, et al. Gemini 1.5: Unlocking multimodal
> > understanding across millions of tokens of context. arXiv preprint arXiv:2403.05530, 2024

---

> > ### Comment · Reviewer_wEH9 · 2025-08-08
> > **Update my score**
> >
> > I have carefully read the authors’ detailed rebuttal and clarifications. I appreciate the explanation of the benchmark’s scale, diversity, and breadth of coverage, as well as its positioning as a unified framework for evaluating a much broader spectrum of table-related tasks than has typically been considered in the ML/NLP community. I also recognize the value of providing a standardized, large-scale testbed that could catalyze future work in this under-explored space, much like [1] and [2] have done for their respective domains.
> >
> >
> > That said, I maintain my earlier view that several of the key empirical findings — such as the performance drop of LLMs on long-context table tasks have been observed in prior work ([3],[4]). While MMTU extends these analyses to a broader set of models and settings, I see these results primarily as confirmations in a new, more comprehensive benchmark rather than entirely novel scientific insights.
> >
> > In my evaluation framework, extending known phenomena to a larger and more diverse setting is valuable but constitutes an incremental rather than a fundamentally novel scientific contribution. Nevertheless, I agree with the authors that the primary novelty of this work lies in constructing a systematic, large-scale benchmark for tables. While the scientific novelty is limited, the benchmark itself represents a significant and timely community resource — much like [1] and [2] — that can enable future research to uncover new patterns. For this reason, I am willing to raise my score.
> >
> > References
> >
> > [1] Hendrycks, Dan, et al. "Measuring massive multitask language understanding." arXiv preprint arXiv:2009.03300 (2020).
> >
> > [2] Yue, Xiang, et al. "Mmmu: A massive multi-discipline multimodal understanding and reasoning benchmark for expert agi." Proceedings of the IEEE/CVF Conference on Computer Vision and Pattern Recognition. 2024.
> >
> > [3] Wang, Lanrui, et al. "NeedleInATable: Exploring Long-Context Capability of Large Language Models towards Long-Structured Tables." arXiv preprint arXiv:2504.06560 (2025).
> >
> > [4] Liu, Tianyang, Fei Wang, and Muhao Chen. "Rethinking tabular data understanding with large language models." arXiv preprint arXiv:2312.16702 (2023).

---

> > > ### Author Response · Authors · 2025-08-08
> > >
> > > Dear Reviewer wEH9,
> > >
> > > Thank you for your thoughtful follow-up, and for recognizing the value of our work. We truly appreciate your willingness to reconsider your score!

---

### Official Review · Reviewer_BU4A · 2025-07-03

**Rating:** 5
**Confidence:** 4

**Summary:**

The authors present MMTU, which is a large-scale benchmark with 30647 questions across 25 real-world table tasks, such as table transform, table matching and table understanding. They benchmarked general and reasoning-focused models and found that the reasoning models perform generally better. In addition, they find that newer models are less sensitive to how tables are formatted. In terms of challenges, LLMs still find long table context or data challenging. This benchmark aims to complement existing reasoning benchmarks by providing tabular data for evaluating tasks that require two-dimensional table understanding, coding, and logical reasoning.

**Dataset Code Accessibility:**

Yes

**Dataset Code Comments:**

The dataset can be found in https://huggingface.co/MMTU-benchmark and the code can be found in https://github.com/MMTU-Benchmark/MMTU

**Ethical Considerations:**

No, there are no or only very minor ethics concerns

**Final Justification:**

The authors responded to my suggestions. I already accepted the paper and have hence, kept my current score!

**Limitations Weaknesses:**

1) The authors already stated the limitations that I was concerned about, namely, the fact that the tasks didn't really other popular table tasks such as table summarization and generation. Their reasoning for including tasks to be objectively evaluated is convincing enough for this work. It would be good to have a good benchmark for these other tasks as part of future work. Perhaps, a suggestion to make them more "objective" is to have heuristics, for instance, on what the "summarized" table information "has" to have.

Clarification questions:

2) To be more specific, what is the exact length considered/tested for long table context (in Figure 6)? The current figure states in percentile, but it would be good to report their exact numbers in the main paper.

3) Figure 9 shows a good observation of a potential "error" by the model, where it uses an abbreviated form of a name instead of the full name. However, could this be due to a naming convention? Does the behavior change or improve when the user explicitly asks for answers to reflect the exact wording used in the data?

**Strengths Contributions:**

1) The paper is well-written and easy to understand. Each of the the data curation workflow steps illustrated in Figure 2 was well-described in Section 3.2 with concise task definitions and examples for table understanding in Table 1 and Figure 1.

2) The tables and figures are informative w.r.t. breakdown of the dataset and how well different models perform on each of these tasks. Tables 2 and Figure 3, for example, give a good overview of what sort of questions to expect in the MMTU benchmark. Figure 5 is a well-constructed performance comparison of all the tested models to get a sense of what each model is good at and what they need to improve on. For instance, most models don't perform as well in KB mapping as compared to table matching, for instance, which through further error analysis, has been found due to hallucinating facts or reciting inaccurate facts.

3) The main reason for the high rating is the overall thoroughness, specially in the data curation pipeline, the clarity around the dataset itself, and the extensive analysis and evaluation involving the data and related models. I also found the benchmarks specification to be well-justified and valuable, and I agree with their design and importance.

---

> ### Author Rebuttal · Authors · 2025-07-31
>
> We are grateful to the reviewer for the recognition and constructive comments. We address the questions raised in the review as follows:
>
> ***It would be good to have a good benchmark for tasks such as table summarization and generation as part of future work. Perhaps, a suggestion to make them more "objective" is to have heuristics, for instance, on what the "summarized" table information "has" to have.***
>
> - We sincerely thank the reviewer for their thoughtful comment and for recognizing our emphasis on objective evaluation criteria. Our decision to focus on objectively evaluatable tasks was made to ensure the reliability and reproducibility of benchmark results, an approach that aligns with the design of other widely adopted and highly popular benchmarks such as MMLU and MMMU.
>
> - We fully agree that extending the benchmark to include more subjective tasks such as table summarization and generation is a valuable future direction. The reviewer’s suggestion to incorporate heuristics (e.g., defining key elements that a summary must contain) to inject partial objectivity is especially insightful. We find this promising and plan to explore such design strategies in future iterations of the benchmark.
>
>
> ***Exact length considered/tested for long table context (in Figure 6)? The current figure states in percentile, but it would be good to report their exact numbers in the main paper.***
>
> - Thank you for pointing this out. In Figure 6, our analysis segments test cases for each task into quartiles based on table length, to enable comparisons across tasks with different length distributions. We agree that reporting the corresponding absolute values would be useful. We have included these numbers below as a markdown table, and will incorporate them into the revised version of the paper.
>
> | task                              | 0–25% range     | 25–50% range    | 50–75% range    | 75–100% range    |
> |:----------------------------------|:----------------|:----------------|:----------------|:-----------------|
> | Arithmetic-Relationship           | 1150.0–3547.0   | 3547.0–6834.0   | 6834.0–15601.5  | 15601.5–126654.0 |
> | Cell-entity-annotation            | 263.0–2138.5    | 2138.5–3706.0   | 3706.0–4693.5   | 4693.5–7526.0    |
> | Column-type-annotation            | 274.0–576.8     | 576.8–1346.5    | 1346.5–3094.8   | 3094.8–6744.0    |
> | Columns-property-anotation        | 261.0–612.8     | 612.8–1416.0    | 1416.0–3004.2   | 3004.2–6754.0    |
> | Data-Imputation                   | 214.0–628.2     | 628.2–1320.0    | 1320.0–1971.0   | 1971.0–102929.0  |
> | Data-transform-pbe                | 262.0–309.8     | 309.8–333.5     | 333.5–412.2     | 412.2–21846.0    |
> | Data-transform-reshape            | 534.0–916.5     | 916.5–1692.0    | 1692.0–3621.0   | 3621.0–76096.0   |
> | Entity-Matching                   | 175.0–207.0     | 207.0–232.0     | 232.0–248.5     | 248.5–425.0      |
> | Error-Detect                      | 211.0–235.0     | 235.0–293.0     | 293.0–638.5     | 638.5–21040.0    |
> | Formula-prediction-context        | 269.0–1096.2    | 1096.2–1315.0   | 1315.0–1564.0   | 1564.0–10332.0   |
> | Functional-Dependency             | 1341.0–3641.0   | 3641.0–5157.0   | 5157.0–7573.0   | 7573.0–76083.0   |
> | List-to-table                     | 184.0–283.5     | 283.5–352.0     | 352.0–514.5     | 514.5–2955.0     |
> | NL2SQL                            | 418.0–872.0     | 872.0–1245.0    | 1245.0–3293.0   | 3293.0–35250.0   |
> | Schema-Matching                   | 547.0–1803.5    | 1803.5–2378.0   | 2378.0–3737.5   | 3737.5–10562.0   |
> | String-Relationship               | 2961.0–7596.2   | 7596.2–14127.5  | 14127.5–26032.5 | 26032.5–92595.0  |
> | Table-Fact-Verification           | 269.0–451.0     | 451.0–584.0     | 584.0–836.5     | 836.5–2959.0     |
> | Table-Locate-by-Row-Col           | 11190.0–21158.0 | 21158.0–22433.0 | 22433.0–23731.0 | 23731.0–72074.0  |
> | Table-QA                          | 229.0–381.0     | 381.0–537.0     | 537.0–847.2     | 847.2–13752.0    |
> | Table-needle-in-a-haystack        | 11123.0–21286.0 | 21286.0–22737.0 | 22737.0–23664.0 | 23664.0–72009.0  |
> | Transform-by-input-output-table   | 200.0–259.0     | 259.0–297.0     | 297.0–366.0     | 366.0–728.0      |
> | Transform-by-output-target-schema | 360.0–1095.5    | 1095.5–1827.5   | 1827.5–2845.8   | 2845.8–113819.0  |
> | equi-join-detect                  | 477.0–2471.0    | 2471.0–3972.0   | 3972.0–6741.0   | 6741.0–107690.0  |
> | header-value-matching             | 186.0–260.0     | 260.0–338.5     | 338.5–489.2     | 489.2–1102.0     |
> | semantic-join                     | 243.0–367.5     | 367.5–742.0     | 742.0–1860.5    | 1860.5–19184.0   |
> | semantic-transform                | 209.0–247.0     | 247.0–265.0     | 265.0–285.0     | 285.0–454.0      |
>
>
> ***Figure 9 shows a good observation of a potential "error" by the model, where it uses an abbreviated form of a name instead of the full name. Could this be due to a naming convention and does the behavior change when the user explicitly asks for answers to reflect the exact wording used in the data?***
>
> - This is an excellent observation. We investigated this case and other similar cases, by prompting models to preserve formatting and naming convention of other values in the same table column, though we observed limited improvement by changing prompts. Our full technical report linked in the paper shows additional screenshots and examples collected from our error analysis (omitted in the submitted version due to space constraints), which reveal similar issues, such as the model failing to count accurately or identify column positions reliably. These behaviors persist even as we adjust prompts. We will highlight these details in our revised paper, which we agree are important for understanding the limitations of current LLMs on table tasks.

---

> > ### Comment · Reviewer_BU4A · 2025-08-05
> >
> > Thank you for the clarification! I'll keep my score.

---

### Official Review · Reviewer_ZZmW · 2025-07-06

**Rating:** 4
**Confidence:** 4

**Summary:**

This paper introduces MMTU, a large-scale benchmark designed to evaluate the table understanding, reasoning, and coding capabilities of large language models (LLMs) on real-world, expert-level tasks. MMTU contains over 30,000 questions across 25 diverse task types, ranging from table transformations and data cleaning to SQL generation and knowledge base mapping. The benchmark has good breadth and depth, drawing tasks from data management and programming languages rather than limiting itself to only NLP domains.

**Dataset Code Accessibility:**

Yes

**Dataset Code Comments:**

Datasets in the submission is readily accessible in a usable format, and well-documented. There is sufficient detail to support reproducibility.

**Ethical Considerations:**

No, there are no or only very minor ethics concerns

**Final Justification:**

Based on the author responses and subsequent discussions, I am maintaining a positive recommendation for this paper. The authors provided thoughtful rebuttals addressing key concerns. Specifically, they clarified that while the benchmark currently emphasizes tasks with objective ground truth for consistency and reliability, they recognize the value of expanding into more subjective task domains. Furthermore, the inclusion of detailed error breakdowns, adversarial stress tests, and empirical checks for data contamination strengthens the benchmark’s credibility and diagnostic value. Overall, the benchmark is a timely and valuable contribution to the community.

**Limitations Weaknesses:**

W1. The benchmark includes only tasks with objective ground truth, which excludes a wide range of subjective or creative tasks like table summarization, synthesis, or enrichment. This may underrepresent challenges found in real-world human-in-the-loop workflows.

W2. The task set is primarily derived from prior research papers, which may miss important industry practices or emerging user needs not yet formalized in academic datasets. This could limit the benchmark’s coverage of some impactful but under-explored scenarios. Hence the realistic natural of the task set is limited.

W3. While the benchmark is challenging, it's unclear if the current formulation supports granular diagnostic evaluation. For example, task-specific breakdowns are helpful, but further fine-grained annotations or adversarial examples could help pinpoint failure modes.\

W4. The paper states that many tasks use existing public datasets. It's not clear whether these benchmarks were used in LLM training. This opens up the possibility that models may have seen parts of the benchmark during pretraining or fine-tuning, especially given the benchmark’s reuse of datasets from older literature.

**Strengths Contributions:**

S1. MMTU is a comprehensive benchmark for evaluating LLMs on table-centric tasks. Unlike prior benchmarks that focus narrowly on tasks like NL-to-SQL or TableQA, MMTU includes 25 different tasks (e.g., data imputation, semantic joins, column transformations), many of which are new to foundation model evaluation. This broader coverage enhances its relevance to real-world professional workflows.

S2. The tasks are well-motivated, grounded in actual user-facing scenarios encountered by data engineers, analysts, and DBAs. The benchmark questions are drawn from 52 real datasets and curated with care to maintain quality and realism, avoiding synthetic or artificially simplified data.

S3. The authors provide a flexible and extensible evaluation framework that goes beyond multiple-choice, supporting structured outputs and code execution for SQL and Python. They also include expert verification of task quality, which adds credibility.

S4. The benchmark results provide valuable insights into model capabilities and limitations. The paper also highlights model fragility with respect to long context and table layout perturbations, which is consistent with the findings in other tasks and is a meaningful contribution to the community's understanding.

---

> ### Author Rebuttal · Authors · 2025-07-31
>
> We sincerely thank the reviewer for your encouragement and insightful feedback. We address the noted weakness as follows:
>
> ***The benchmark includes only tasks with objective ground truth***
>
> - We appreciate the reviewer raising this important point. Our benchmark currently focuses on tasks with objective ground truth, which covers a wide range of practical scenarios where users interact with tables to accomplish tasks, using code and other means. This design choice ensures reliable and reproducible evaluation, minimizing reliance on subjective assessments or LLM judges. We should highlight that this approach aligns with other widely adopted and highly popular benchmark efforts, such as MMLU, and MMMU, which similarly use tasks with deterministic answers (e.g., multiple-choice questions) to enable consistent comparisons across models.
>
> - We fully agree that expanding the benchmark to include tasks without objective ground truth, such as table summarization, is a valuable direction. We appreciate the suggestion and plan to explore such directions in future iterations.
>
>
> ***The task set is primarily derived from prior research papers, which may miss important industry practices***.
>
> - Thank you for this helpful observation. While our tasks are indeed sourced from the academic literature, many of the tasks actually originate from studies by industry researchers based on real-world user needs (e.g., [86, 35, 124, 96, 88, 135, 87, 116, 45, 65, 62, 106, 49]). These works typically employ real datasets and real user queries/tasks collected from the wild or from production settings, thus reflecting industry-relevant challenges. We will make this clearer in the revised version of the paper by highlighting the industrial origins and practical relevance of these tasks.
>
>
> ***Support of granular diagnostic evaluation***.
>
> - This is a great point. We indeed provide fine-grained annotations of failure modes in Section 4.3 of our paper, using a detailed categorization of errors, such as (1) table understanding errors, (2) reasoning and coding errors, (3) knowledge-based errors, etc., which help pinpoint LLM limitations on table tasks, and point to directions for future improvements.
> - Moreover, we include adversarial settings (e.g., long-context tables) and detailed case studies, such as the Needle in a Haystack task (Appendix D), to stress test model robustness. We acknowledge that some of these results are currently buried in the appendix due to space constraints, which we will highlight more in our revised paper.
>
>
>
> ***Possible data contamination as these benchmarks may be used in LLM training***.
>
> - We acknowledge the general concern regarding potential data contamination in LLM evaluation. However, several factors indicate that our benchmarks are unlikely to have been memorized by current LLMs:
>
>   - (1)  Low model performance: Even state-of-the-art reasoning models achieve relatively low accuracy (e.g., ~0.6), suggesting that the models are not familiar with the tasks or data.
>
>
>   - (2) Multimodal and fragmented inputs: Our benchmarks require reasoning across multiple modalities (e.g., large tables, code snippets, ground-truth row IDs), often distributed across separate files. This structure is fundamentally different from typical NLP benchmarks that involve single, contiguous text inputs and are more prone to memorization.
>
>
>   - (3) Limited data accessibility: Many datasets used in our benchmark are relatively obscure, or deeply buried within supplementary materials, multi-file folders, and coded database files.
>
>
>   - (4) Empirical tests for memorization: When we query leading LLMs using prefix fragments from our benchmarks, they neither recognize the prompts nor reproduce the ground-truth answers, providing further empirical evidence against memorization.
>
> We hope these address the concern and reinforce the benchmark’s validity for robust model evaluation.

---

> > ### Comment · Reviewer_ZZmW · 2025-08-05
> >
> > Thank you for your detailed responses and clarification. I'll keep my score given it's already positive.

---

### Note · Authors · 2025-08-14

We sincerely thank all reviewers for their thoughtful engagement and feedback throughout the discussion phase, and especially the Area Chair for facilitating such a productive dialogue. We are truly grateful to Reviewers **BU4A** and **ZZmW** for their encouragement and positive evaluations from the start, which affirmed the value of our work. We also deeply appreciate Reviewers **BG3j** and **wEH9** for engaging in in-depth discussions that clarified key points and drove substantial improvements, leading to positive re-assessments of their initial scores.

The feedback we received has already led to significant enhancements to our work, including:
- **Enhanced LLM-based ambiguity and quality filtering** to further improve dataset rigor.


- **Expanded model coverage** to include 14 additional chat and reasoning models, enabling more comprehensive and balanced comparisons.


- **Improved evaluation using extensibility**, such as addressing order-sensitivity in execution accuracy, which further demonstrated the extensibility of our evaluation framework.


- **Strengthened error analysis** using a fine-grained taxonomy to categorize errors, with detailed distributions and illustrative examples.


We are pleased that reviewers recognize the value of MMTU as a **timely, large-scale, and comprehensive benchmark for table-centric tasks**, and its potential to catalyze future research – much like MMLU and MMMU have done in their respective domains. We are grateful for the opportunity to improve our work through this process, and we look forward to releasing the enhanced benchmark to support and inspire future breakthroughs in the area of table understanding and reasoning.

---

### Decision · Program_Chairs · 2025-09-18

**Decision:**

Accept (poster)

**Comment:**

This paper present MMTU, a large-scale benchmark for evaluating LLM over table understanding and reasoning tasks. Reviewers advocated the paper for its comprehensive/unified scope with well-motivated scenarios and well-thought out framework and the detailed analysis that revealed valuable insights. They also identified some areas for improvements, which were mostly clarified during rebuttal, although there were some remaining concerns that could be improved.